# Gut bacteria induce oviposition preference through ovipositor recognition in fruit fly

Muyang He[1], Huimin Chen[1], Xiaorui Yang[1], Yang Gao[1], Yongyue Lu [1✉] & Daifeng Cheng [1✉]

Gut bacteria play important roles in insect life cycle, and various routes can be used by insects to effectively transmit their gut bacteria. However, it is unclear if the gut bacteria can spread by actively attracting their insect hosts, and the recognition mechanisms of host insects are poorly understood. Here, we explore chemical interactions between *Bactrocera dorsalis* and its gut bacterium *Citrobacter* sp. (CF-BD). We found that CF-BD could affect the development of host ovaries and could be vertically transmitted via host oviposition. CF-BD could attract *B. dorsalis* to lay eggs by producing 3-hexenyl acetate (3-HA) in fruits that were hosts of *B. dorsalis*. Furthermore, we found that *B. dorsalis* could directly recognize CF-BD in fruits with their ovipositors in which olfactory genes were expressed to bind 3-HA. This work reports an important mechanism concerning the active spread of gut bacteria in their host insects.

[1] Department of Entomology, South China Agricultural University, Guangzhou 510640, China. ✉email: luyongyue@scau.edu.cn; chengdaifeng@scau.edu.cn

Since the offspring lack the care of their mothers, many insects will find the best oviposition site to lay eggs[1]. Many factors can influence the oviposition selection of insects, such as the environment, natural enemies, conspecific or heterogeneous individuals and host plants[2,3]. Recent years, many studies have even indicated that insect reproduction and behavior can be manipulated by their symbionts[4,5]. Insects may use their olfaction, gustation, vision and tactile sense systems to receive information on egg-laying sites[6,7].

Chemical volatiles coming from multiple sources (such as host plants, natural enemies and conspecifics) act as important information that can affect the oviposition selection of insects. For instance, the adjustment of oviposition strategy is an important part of the response of herbivorous insects against the defensive response of host plants[8]. Natural enemies can also affect the egg-laying decision by emitting signals. Many species of mosquitoes, for example, can recognize kairomone released by their predators and avoid laying their eggs[9,10]. To reduce competitive pressure, insects can avoid laying eggs at sites with conspecifics by identifying the volatiles released by conspecifics[8]. In recent decades, an increasing number of studies on symbiotic microbes have indicated that insect behaviors, including egg-laying, can be regulated by symbionts in multiple ways. For example, *Bactrocera oleae* that have native gut-symbionts attempted oviposition significantly more times than axenic flies[11]. Foraging behavior of flies can also be affected by the intestinal bacteria[12–15]. Specific bacteria-associated carboxylic acids and methyl esters can strongly stimulate *Aedes aegypti* to lay eggs[16]. Moreover, aversive responses of insects can also be induced by intestinal bacteria[17,18].

Gut bacteria are widespread in almost all insects[19], and the relationships between gut bacteria and insects have partly resulted in the diversification and evolutionary success of insects[20]. To date, the known roles of gut bacteria in insect hosts include colonization resistance against pathogens or parasites[21], intestinal cell renewal and promotion of systemic growth[22], degradation of cellulose[23], degradation of toxins ingested with the diet[24], nutrient supplementation[25] and intraspecific or interspecific communication[26,27]. To transmit gut bacteria, some gregarious insects have evolved specific mechanisms for bacterial transfer to progeny, such as egg smearing[28] or egg capsules[29]. However, many insects lack reliable mechanisms for the direct transmission of gut bacteria. For the benefit of gut bacteria, attracting host insects to lay eggs may ensure efficient transmission in host populations. However, there are few evolutionary neurobiology studies on how insects recognize their gut bacteria[30,31].

Recently, we have isolated the gut bacterium *Citrobacter* sp. (CF-BD) from *Bactrocera dorsalis*. We have demonstrated that CF-BD is capable of degrading trichlorphon and thus increasing host trichlorphon resistance[32]. CF-BD can be vertically transmitted to the eggs and has a significant effect on the fecundity of *B. dorsalis*[33], which suggests the important roles of CF-BD for *B. dorsalis*. In this study, we further demonstrated that CF-BD can improve the oviposition preference of *B. dorsalis* by producing 3-HA in fruits. Moreover, gravid females can sense 3-HA in fruits with their ovipositors. This work suggests an efficient interaction mechanism between *B. dorsalis* and CF-BD.

## Results

### CF-BD can be transmitted into *B. dorsalis* ovaries and are associated with ovary development

CF-BD was located in females with fluorescence in situ hybridization (FISH). FISH results indicated that CF-BD was present in mature female ovary while absent in newly emerged female ovary (Fig. 1a, b), indicating that CF-BD can enter into the ovary at the mature stage.

Moreover, histological sections and staining results showed that CF-BD can be detected in dissected rectums and ovaries of mature females (Fig. 1c, d). Nested PCR results also indicated that 100% ovaries (19/19) of mature females carried CF-BD (Fig. 1e). To verify the influence of CF-BD on ovary development, streptomycin (CF-BD is sensitive to streptomycin[32]) or CF-BD suspension was injected in the abdomen of newly emerged flies. And the development of ovaries was significantly inhibited or accelerated by injecting streptomycin or CF-BD into the abdomen of the females (Fig. 1f, g: Width: $n \geq 13$ replicates, $F = 8.158$, $df = 3$, 53, $P = 0.0001$; Length: $n \geq 13$ replicates, $F = 15.31$, $df = 3$, 53, $P < 0.0001$; Fig. 1h, i: Width: $n \geq 13$ replicates, $F = 13.01$, $df = 3$, 64, $P < 0.0001$; Length: $n \geq 13$ replicates, $F = 15.54$, $df = 3$, 53, $P < 0.0001$. Data was analyzed by ANOVA followed by Tukey's test). Given that previous study showed that CF-BD could be transmitted into fruits by egg-laying and reproduce rapidly in fruits[33], we infer that the reproduced CF-BD may attract females to lay eggs in CF-BD-infected fruits to ensure that CF-BD can be effectively transmitted in the population of *B. dorsalis*.

### Females show oviposition preference to CF-BD mixed puree

To verify the above hypothesis, we tested *B. dorsalis* oviposition preference in a 2-choice assay for which the flies were allowed to oviposit in either CF-BD mixed purees or the control purees (Fig. 2a). The results showed that the females prefer lay eggs in CF-BD mixed purees after 4 h (Fig. 2b: Mango, 0 h: $n = 10$ replicates, $P = 0.625$, Sum of signed ranks (W) $= -11$; 4 h: $n = 10$ replicates, $P = 0.0098$, Sum of signed ranks (W) $= -49$; Guava, 0 h: $n = 10$ replicates, $P = 0.6953$, Sum of signed ranks (W) $= -9$; 4 h: $n = 10$ replicates, $P = 0.002$, Sum of signed ranks (W) $= -55$. Data was analyzed by Wilcoxon matched-pairs signed rank test), which indicated that the chemicals that attract oviposition may be produced by CF-BD after 4 h. To identify the active compounds responsible for oviposition attraction, the volatiles in the purees were collected with solid phase microextraction (SPME) and identified by Gas Chromatography-Mass Spectrometer (GC–MS). The results indicated that 3-HA was produced in mango purees mixed with CF-BD after 4 h (Fig. 2c). Although 3-HA was identified in the two types of guava purees (Fig. 2d), the 3-HA content in purees mixed with CF-BD was significantly higher (Fig. 2e: $n = 5$ replicates, $t = 5.882$, $P = 0.0042$. Data was analyzed by paired sample student *t*-test). Moreover, an oviposition preference comparison between purees and purees mixed with 3-HA showed that females preferred to lay eggs in purees mixed with 3-HA (Fig. 2f: Mango: $n = 15$ replicates, Kendall W $= 0.3067$, $\chi^2 = 13.8$, $P = 0.003$; Guava: $n = 11$ replicates, Kendall W $= 0.365$, $\chi^2 = 12.05$, $P = 0.007$. Data was analyzed by the Kendall nonparametric test. Also see Supplementary Fig. 1). These results indicate that more 3-HA produced by CF-BD is the volatile responsible for oviposition attraction.

### *B. dorsalis* preference for CF-BD mixed purees is limited to oviposition

To test whether gravid females use their antennae to sense 3-HA in the oviposition site, we conducted olfactory trap assays in a transparent cage. In the cage, two vials contained purees and CF-BD added purees were placed to capture the gravid females (Fig. 3a). Such device can make sure that a relatively long distance exists between the purees and the females. Then the females could only recognize the puree in the vial with their antennae. However, we found that gravid females were not significantly attracted by purees mixed with CF-BD (Fig. 3b: Mango, 0 h: $n = 10$ replicates, $P = 0.1563$, Sum of signed ranks (W) $= 25$; 4 h: $n = 10$ replicates, $P = 0.3223$, Sum of signed ranks (W) $= 21$; Guava, 0 h: $n = 10$ replicates, $P = 0.2207$, Sum of signed ranks (W) $= 25$; 4 h: $n = 10$ replicates, $P = 0.9766$, Sum of

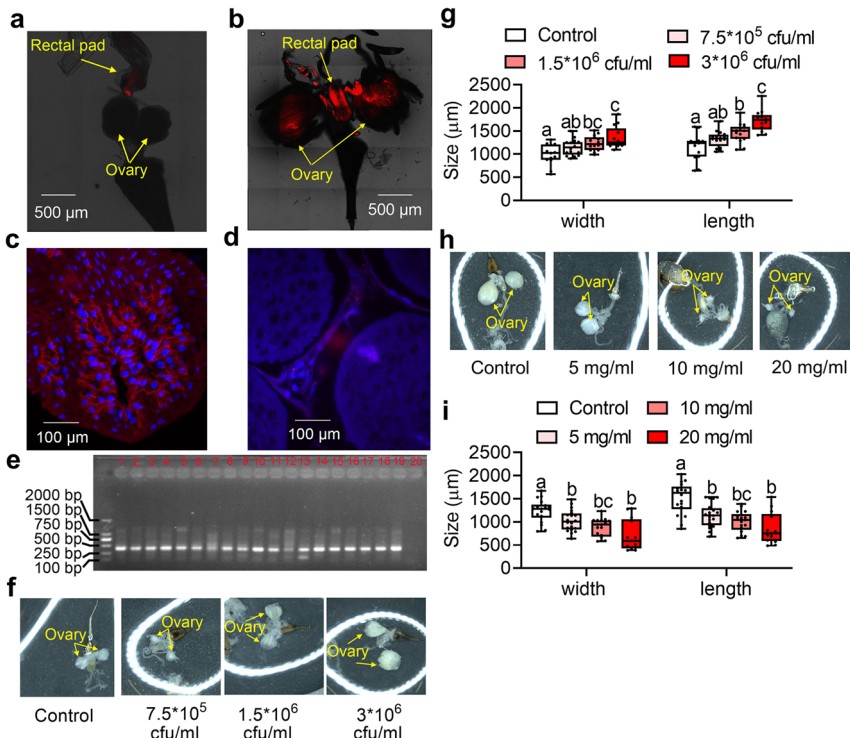

**Fig. 1 CF-BD localization and its effect on ovary development. a**, **b** CF-BD localization in the rectum and ovary of newly emerged female and mature female. The red signals indicate CF-BD. Scale bars are 500 μm. **c**, **d** CF-BD localization in the rectum and ovary of mature female with histological sections and staining. The red signals indicate CF-BD. Scale bars are 100 μm. **e** CF-BD identification in ovaries of mature females with nested PCR. PCR amplification product size of CF-BD is 371 bp. Lane 1–19: ovary samples, lane 20: negative control. **f**, **g** Box-and-whisker plots show ovary width and length of female injected with CF-BD and control. **h**, **i** Box-and-whisker plots show ovary width and length of female injected with streptomycin and control. In box-and-whisker plots, where the boxes encompass the first to the third quartiles, inside the box the horizontal line shows the median, and the whiskers are the maximum and minimum observation. Different letters above the error bars indicate significant differences at the 0.05 level analyzed by ANOVA followed by Tukey's test. Detail information for sample sizes and statistics can be found in Supplementary Data 2.

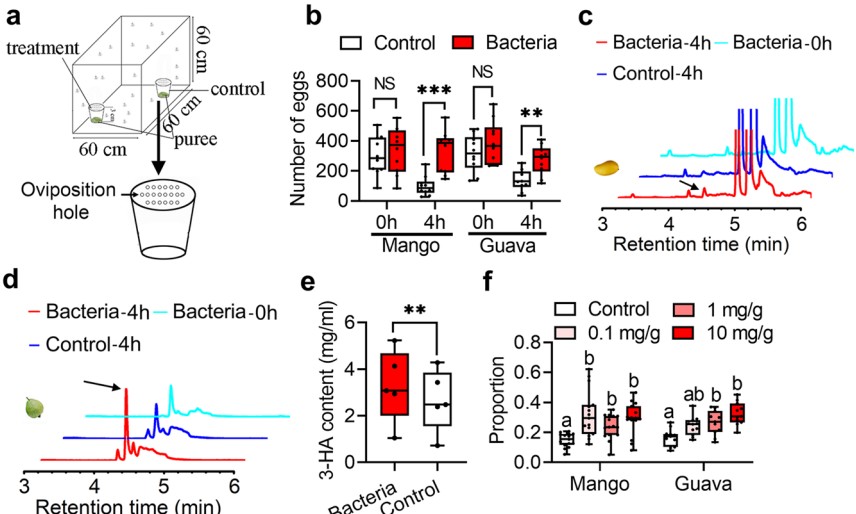

**Fig. 2 Oviposition attraction of CF-BD to *B. dorsalis*. a** Device used to test oviposition preference. **b** Box-and-whisker plots show oviposition attraction of mango and guava purees mixed with CF-BD at different times. Data was analyzed by Wilcoxon matched-pairs signed rank test. "NS" no significance, **P < 0.01, ***P < 0.001. **c**, **d** Volatile identification in mango and guava purees added with CF-BD. The black arrows indicate 3-HA. **e** Box-and-whisker plots show 3-HA content comparison between CF-BD added guava puree and control. Data was analyzed by paired sample student *t*-test. **P < 0.01. **f** Box-and-whisker plots show egg proportions comparison between purees and purees supplemented with 3-HA. Different letters above the error bars indicate significant differences at the 0.05 level with the Kendall nonparametric test. In box-and-whisker plots, where the boxes encompass the first to the third quartiles, inside the box the horizontal line shows the median, and the whiskers are the maximum and minimum observation. Detail information for sample sizes and statistics can be found in Supplementary Data 2.

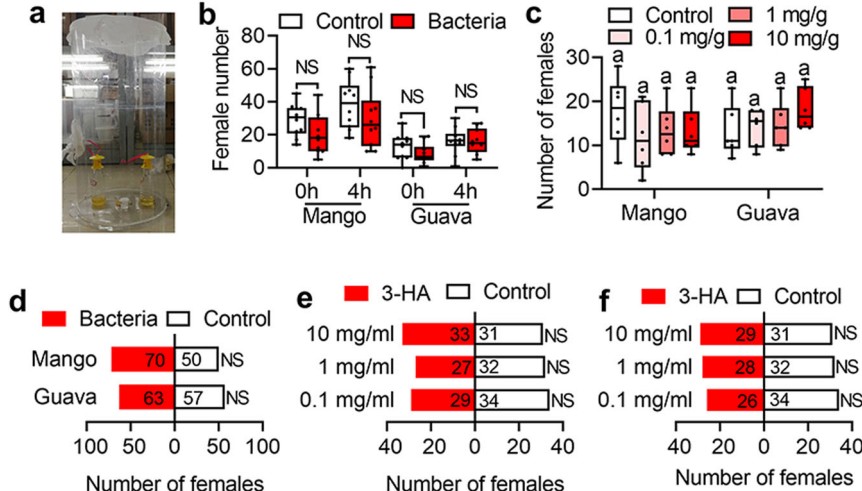

**Fig. 3 Olfactory trap effect of CF-BD and 3-HA. a** A transparent cage used to capture the female flies. **b** Box-and-whisker plots show olfactory attraction of mango and guava purees added with CF-BD at different times. Data was analyzed by Wilcoxon matched-pairs signed rank test. "NS" no significant difference. **c** Box-and-whisker plots show female number comparison between purees and purees supplemented with 3-HA. The same letters above the error bars indicate no significant differences at the 0.05 level with the Kendall nonparametric test. **d** Attraction effect of purees added with CF-BD to females. Fly numbers are shown in the bars. Data was analyzed by chi-square test. "NS" no significance. **e**, **f** Attraction effect of mango and guava purees added with 3-HA to females. Fly numbers are shown in the bars. Data was analyzed by chi-square test. "NS" no significant difference. In box-and-whisker plots, where the boxes encompass the first to the third quartiles, inside the box the horizontal line shows the median, and the whiskers are the maximum and minimum observation. Detail information for sample sizes and statistics can be found in Supplementary Data 2.

signed ranks (W) = 1. Data was analyzed by Wilcoxon matched-pairs signed rank test). Purees mixed with 3-HA could not significantly attract gravid females either (Fig. 3c: Mango: $n = 6$ replicates, Kendall W = 0.099, $\chi^2 = 1.78$, $P = 0.619$; Guava: $n = 6$ replicates, Kendall $W = 0.1$, $\chi^2 = 1.8$, $P = 0.615$. Data was analyzed by the Kendall nonparametric test). Individual choice assays in a Y shape olfactometer also showed that gravid females did not show preference for purees mixed with CF-BD or 3-HA (Fig. 3d: Mango: $\chi^2 = 0.3$, $P = 0.584$; Guava: $\chi^2 = 3.333$, $P = 0.068$; Fig. 3e: 0.1 mg/ml: $\chi^2 = 0.397$, $P = 0.529$; 1 mg/ml: $\chi^2 = 0.424$, $P = 0.515$; 10 mg/ml: $\chi^2 = 0.063$, $P = 0.803$; Fig. 3f: 0.1 mg/ml: $\chi^2 = 1.067$, $P = 0.302$; 1 mg/ml: $\chi^2 = 0.267$, $P = 0.606$; 10 mg/ml: $\chi^2 = 0.067$, $P = 0.796$. Data was analyzed by chi-square test). These results indicate that purees mixed with CF-BD couldn't attract gravid females at a relatively long distance. The reason why females prefer to lay eggs in purees mixed with CF-BD may be that the females can only use their ovipositors to recognize 3-HA in purees mixed with CF-BD at a close distance. To verify such hypothesis, egg laying behavior was recorded by a camera. The video showed that the ovipositor of a female would frequently extend out to receive the chemical information if the females were close to the puree (Supplementary movie 1). Thus, we inferred that the oviposition preference to purees mixed with CF-BD may be regulated by the olfactory genes expressed in the ovipositor.

**Ovipositors expressing olfactory genes can recognize 3-HA.** To test the above hypothesis, scanning electron microscopy was used to identify the sensilla on the ovipositor. The scanning results indicated a single type of coeloconic sensilla identified on the ovipositor (Fig. 4a). Thus, we further tested the electroantennography (EAG) response of the ovipositor with 3-HA as the stimulant. The results showed that 3-HA could trigger a marked EAG response in ovipositors (Fig. 4b: $n = 10$ replicates, $F = 12.71$, $df = 3, 36$, $P < 0.0001$. Data was analyzed by ANOVA followed by Tukey's test). To further identify whether the olfactory genes were expressed and play roles in receiving 3-HA in the ovipositor, ovipositor comparative transcriptome analysis was performed for ovipositors at different developmental times. The results showed that the expression of 754

differentially expressed genes exhibited an upward trend in ovipositors from eclosion to sexual maturity (Supplementary Fig. 2, Supplementary Data 1). By screening the olfactory genes among the above genes, 5 odorant binding protein genes (*Obp*) were identified (Fig. 4c). 1 odorant binding protein gene (*Obp56d-2*) highly expressed in the mature stages of ovipositors was also investigated in our research (Fig. 4c). Besides, one odorant receptor gene (*OR7a-2*) and 1 ionotropic receptor (*IR25a*) were also included in our study since they were the only identified *OR* and *IR* gene in the ovipositors (Fig. 4c). We further verified the expression profiles of the above genes in the ovipositor and antenna by qRT-PCR. The results showed that *Obp56d* and *Obp56d-2* had the highest expression level in the ovipositor of the mature females (Fig. 4e: $n = 3$ replicates, $F = 16.2$, $df = 5, 12$, $P < 0.0001$; Fig. 4f: $n = 3$ replicates, $F = 25.74$, $df = 5, 12$, $P < 0.0001$. Data was analyzed by ANOVA followed by Tukey's test). Although *Obp19d*, *OR7a-2* and *IR25a* were also expressed in ovipositors, their expression levels were much lower than those in antennae (Fig. 4d: $n = 3$ replicates, $F = 16.05$, $df = 5, 12$, $P < 0.0001$; Fig. 4g: $n = 3$ replicates, $F = 19.93$, $df = 5, 12$, $P < 0.0001$; Fig. 4h: $n = 3$ replicates, $F = 103.6$, $df = 5, 12$, $P < 0.0001$. Data was analyzed by ANOVA followed by Tukey's test). Since the above results indicated that ovipositors are likely to be used to sense 3-HA, we infer ovipositor specifically expressed Obp56d and Obp56d-2 may play roles in binding 3-HA.

**Ovipositor specifically expressed olfactory genes play roles in regulating oviposition preference.** To verify the function of the identified olfactory genes, we tested the ability of *Obp56d* and *Obp56d-2* to bind 3-HA with the competitive binding assays in vitro. SDS-PAGE showed that the molecular weight of the recombinant proteins of Obp56d and Obp56d-2 were about 29 kDa, which was in line with the expected molecular weight of the target proteins (Supplementary Fig. 3a). Binding assays showed that Obp56d and Obp56d-2 could effectively bind with 1-NPN (Fig. 5a). However, competitive binding experiments showed that 3-HA can reduce the relative fluorescence intensity of the Obp/1-NPN complex by >50% (Fig. 5b), indicating that Obp56d and Obp56d-2 had strong binding abilities to 3-HA. Then we further

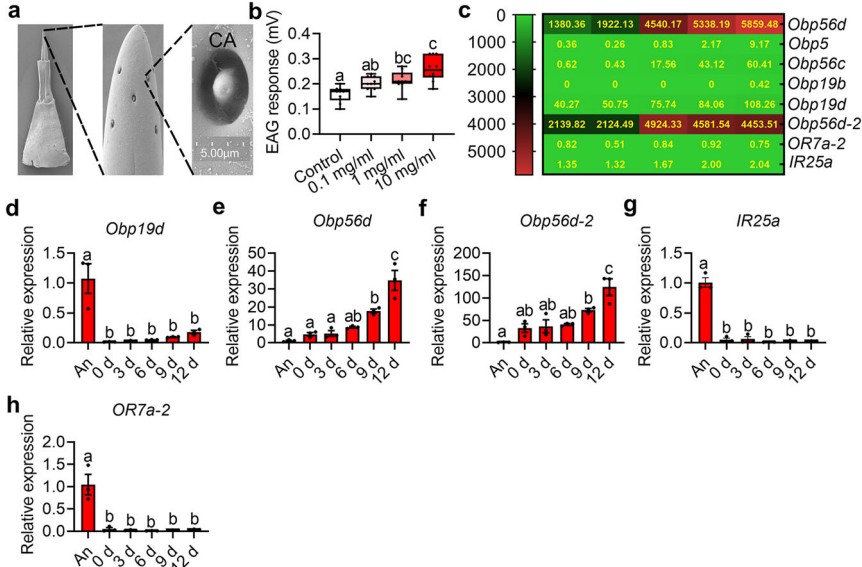

**Fig. 4 Function of ovipositors as receptors for 3-HA and olfactory gene identification. a** Sensilla scanning on the ovipositor. CA: coeloconic sensilla. Scale bars are 5 μm. **b** Box-and-whisker plots show EAG response of the ovipositors to 3-HA. Different letters above the error bars indicate significant differences at the 0.05 level by ANOVA followed by Tukey's test. **c** Expression patterns of the identified olfactory genes in ovipositors at different developmental times. **d–h** Box-and-whisker plots show relative expression of olfactory genes in antennae and ovipositors. An: antennae. Different letters above the error bars indicate significant differences at the 0.05 level by ANOVA followed by Tukey's test. Data in bar plots show mean values ± SEM. In box-and-whisker plots, where the boxes encompass the first to the third quartiles, inside the box the horizontal line shows the median, and the whiskers are the maximum and minimum observation. Detail information for sample sizes and statistics can be found in Supplementary Data 2.

investigated the roles of the identified olfactory genes in regulating oviposition preference with RNAi. For the *Obp*, the expression of *Obp56d* and *Obp56d-2* were significantly down regulated after dsRNA injection (Supplementary Fig. 3b, c). However, the oviposition preference index was not significantly decreased with either *Obp56d* or *Obp56d-2* being knocked down; Interestingly, the index decreased significantly when both *Obp56d* and *Obp56d-2* were knocked down (Fig. 5c: $n = 9$ replicates, Kruskal–Wallis statistic $= 13.67$, $P = 0.0084$. Data was analyzed by Kruskal–Wallis test followed by Dunn's test. Also see Supplementary Fig. 4). Moreover, ovipositor EAG response to 3-HA of the flies with both *Obp56d* and *Obp56d-2* being knocked down decreased significantly (Fig. 5d: $n = 3$ replicates, $F = 26.99$, $df = 2, 6$, $P = 0.001$. Data was analyzed by ANOVA followed by Tukey's test). These results indicate that an olfactory pathway exist in the ovipositor and play roles in regulating oviposition preference by recognizing 3-HA. To further identify the potential odorant receptor of 3-HA, we also measured the oviposition preference index with *OR7a-2* and *IR25a* being knocked down (Supplementary Fig. 3d, e). The oviposition preference for 3-HA could not be decreased with either *OR7a-2* or *IR25a* being knocked down (Fig. 5e: $n \geq 8$ replicates, $F = 0.6632$, $df = 2, 22$, $P = 0.5252$; Fig. 5f: $F = 0.2428$, $df = 2, 16$, $P = 0.7873$. Data was analyzed by ANOVA followed by Tukey's test. Also see Supplementary Fig. 5).

## Discussion
This study indicates that oviposition attraction to vertically transmitted CF-BD evolved in *B. dorsalis*. Females can transmit CF-BD from gut to ovary, and then CF-BD can be delivered into fruits by laying eggs. More interestingly, females can recognize 3-HA in CF-BD-colonized fruits through olfactory genes in their ovipositor.

Female insects select oviposition sites that maximize the survival, growth, and reproductive potential of the offspring[7,34,35]. Ovipositing in a fruit that is occupied by other larvae can be maladaptive for female flies, and indeed B. *dorsalis* females prefer

intact fruit over rotting fruit[18,36]. Furthermore, our previous study indicated that egg-surface bacteria (such as *Providencia* sp. and *Klebsiella* sp.) can deter females from ovipositing[17]. However, the current study indicates that females of *B. dorsalis* prefer to lay eggs in CF-BD infected fruits, contradicts the previous study. *B. dorsalis* females will usually lay dozens of eggs in one fruit and clutch size varies greatly, according to host density[37]. Females will tend to distribute eggs more widely and reduce the number of eggs laid per oviposition bout under high host density[37]. Thus, host density may be the reason to explain the contradiction: when hosts are scarce, competition is important and it is adaptive to repel conspecifics from an oviposition site. This may also be true for host species that are limited in resources (e.g. very small fruits). On the other hand, when host resources are abundant (many hosts or large fruits), it may be advantageous to have many larvae inside each host, as the action of many larvae and the larvae carried bacteria can make nutritional resources available to all the larvae in the fruit, and favor optimal development. And an alternative hypothesis is that CF-BD may be present in the environment and establish on suitable hosts, which become more attractive to oviposition as a result. Furthermore, more investigations should be done in the future to reveal whether there actually is an advantage in recruiting more oviposition attempts to a host.

*Citrobacter* is ubiquitous in nature can be isolated from many fruits in orchards[38–40]. Some strains of *Citrobacter* can even produce ethanol efficiently using fruit waste[41]. A number of studies have indicated that *Citrobacter* plays important nutrition supplying roles in both larva and adult of *Bactrocera*[42–44]. Thus, the evolution of symbiotic interactions between *Citrobacter* and *B. dorsalis* may have been initiated by commensal or facultatively fruit pathogenic *Citrobacter* that exploit compounds present in the fruits that are hosts of *B. dorsalis*. Once *Citrobacter* is fed by *B. dorsalis*, *Citrobacter* can help to digest fruit substances and make fruit substances being easily absorbed by larvae. The bacterial-vector relationship may accelerate dissemination of *Citrobacter* in

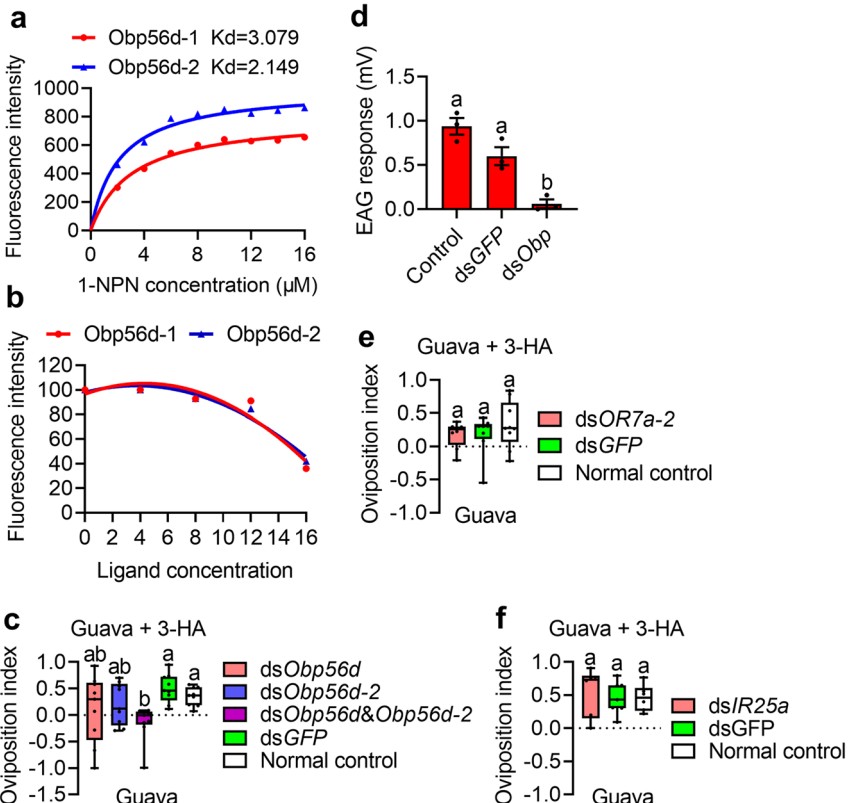

**Fig. 5 Olfactory gene function verification. a** Binding abilities of 1-NPN to Obp56d and Obp56d-2. **b** Competitive binding ability of 3-HA to 1-NPN. **c** Box-and-whisker plots show oviposition preference of females for 3-HA with *Obp* genes knocked down. Different letters above the error bars indicate significant differences at the 0.05 level by Kruskal–Wallis test followed by Dunn's test. **d** Box-and-whisker plots show ovipositor EAG response to 3-HA (10 mg/ml) with both *Obp56d* and *Obp56d-2* being knocked down. Different letters above the error bars indicate significant differences at the 0.05 level by ANOVA followed by Tukey's test. Data in bar plots show mean values ± SEM. **e** Box-and-whisker plots show oviposition preference of females for 3-HA with *OR7a-2* knocked down. The same letters above the error bars indicate no significant differences at the 0.05 level by ANOVA followed by Tukey's test. **f** Box-and-whisker plots show oviposition preference of females for 3-HA with *IR25a* knockdown. The same letters above the error bars indicate no significant differences at the 0.05 level by ANOVA followed by Tukey's test. In box-and-whisker plots, where the boxes encompass the first to the third quartiles, inside the box the horizontal line shows the median, and the whiskers are the maximum and minimum observation. Detail information for sample sizes and statistics can be found in Supplementary Data 2.

fruits as that in other insect-phytopathogens[45]. As a result, *Citrobacter* can provide possibly digestion (or enrichment of dietary nutrients), defense against detrimental microbes and adults ovary development to *B. dorsalis*. *Citrobacter* can be transmitted to new habitats by attracting oviposition and spreading with mature adults.

Although CF-BD can be acquired by larvae through oviposition, the amount acquired in this way is relatively low and may be insufficient to play a corresponding role, especially in the face of complex wild habitats. Thus, the mother may seek fruits with more CF-BD colonization to lay their eggs to ensure that the offspring can acquire enough CF-BD after hatching. Consistent with this hypothesis, we found that puree supplemented with CF-BD could produce 3-HA to attract gravid females to lay eggs. However, 3-HA added to the purees did not perfectly recover the egg-laying attraction effect of CF-BD to *B. dorsalis*, which might be explained by the fact that 3-HA was not the only volatile attracting oviposition, and GS-MS analysis could not identify the other volatiles with subtle differences between CF-BD mixed purees and the control. Given that studies have indicated that attraction or aggregation pheromones can be produced by bacteria[46], yeast[47], fungi[48] and even pathogens[49] of insects, regulation of insect behavior by volatiles produced by microorganisms may be widespread. Though there are numerous vertical transmitted symbionts in *B. dorsalis*, the oviposition attraction

properties should be specific to CF-BD. In our previous study, we have shown that the dominant vertical transmitted bacteria didn't have the properties of CF-BD in this study[17]. Most importantly, our study have shown that there was no 3-HA production if all the bacteria get into the fruit via oviposition[17]. In this regard, it should be considered that only CF-BD has the property to produce 3-HA and attract oviposition.

For female insects, finding suitable habitats is very important for the survival of their offspring, and the olfactory system plays a key role in such processes[50]. Although antennae are often considered to be the nostrils of insects[51], some species also use other organs, such as maxillary palps and proboscis, to detect volatile compounds[52,53]. Based on behavior and molecular biology assays, we found that oviposition attraction to 3-HA was regulated by the ovipositor specifically expressed olfactory genes in *B. dorsalis*. To the best of our knowledge, such results are reported in fruit flies for the first time. Usually, one Obp is expressed to bind the volatile and deliver it to the OR[54–56]. In our study, we have found both *Obp* (*Obp56d* and *Obp56d-2*) genes played roles in binding 3-HA. Knocking down either *Obp56d* or *Obp56d-2* did not suppress the oviposition preference for 3-HA, while the preference disappeared when both *Obp56d* and *OBP56d-2* were knocked down. Previous studies in aphids also indicated that more than one Obp can bind alarm pheromones[57,58]. For aphids, sensing alarm pheromones or not means life or death. More than one *Obp*

expressed to bind alarm pheromones may increase the binding efficiency and increase the survival rate of aphids facing danger. For *B. dorsalis*, more *Obp* expressed to bind 3-HA may also increase the efficiency of receiving CF-BD for offspring. The odorant receptors (OR or IR) expressed in olfactory neurons are vital to receive volatiles delivered by the Obp[59,60]. Recent studies in moths have found that ORs, IRs and their coreceptors are also expressed in ovipositors, and ovipositors perform the same odor-recognition function as antennae[61,62]. In our study, only one OR (OR7a-2) (no coreceptors) and one IR (IR25a) were identified in the ovipositor. Knocking down any one of the genes could not suppress the oviposition preference. However, only coeloconic sensilla can be observed on *B. dorsalis* ovipositors, and it is generally thought that neurons in coeloconic sensilla mainly express IR[60,63]. Thus, IR expressed in the coeloconic sensilla might play roles in sensing 3-HA and then regulating oviposition preference. We need to further identify the IRs playing roles in sensing 3-HA in the ovipositor in the future.

This work reveals a mechanism concerning the transmission of gut bacteria in insects. Previous studies have indicated that a series of extracellular transmission routes for bacterial symbionts (such as environmental determination[64], coprophagy[65], smearing of brood cells or egg surfaces[28], social acquisition[66], capsule transmission or infection via jelly like secretions[29,67]) are applied by insects. Our study indicates that a more complex transmission route is applied by *B. dorsalis* to efficiently acquire gut bacteria. The mother fly can vertically transmit gut bacteria to larvae by laying egg and gut bacteria can attract other mother flies to lay eggs by producing volatiles with fermentation in the host fruits of the flies. With such route, gut bacteria can be efficiently transmitted in the fly population.

## Material and methods

**Insect rearing.** The *B. dorsalis* strain collected from a carambola (*Averrhoa carambola*) orchard in Guangzhou, Guangdong Province, was reared under laboratory conditions (27 ± 1 °C, 12:12 h light:dark cycle, 70–80% RH). A maize-based artificial diet containing 150 g of corn flour, 150 g of banana, 0.6 g of sodium benzoate, 30 g of yeast, 30 g of sucrose, 30 g of paper towel, 1.2 mL of hydrochloric acid and 300 mL of water was used to feed the larvae. Adults were fed a solid diet (consisting of 50 g yeast and 50 g sugar) and 50 mL sterile water in a 35 cm × 35 cm × 35 cm wooden cage. For *B. dorsalis*, the female will start laying eggs once mated and the female will start mating 7 days after emergence. To make sure all females used in our study were gravid females, females were selected 10 day after emergence.

**Visualization of CF-BD with FISH and PCR.** FISH was carried out on dissected gut and ovary samples from *B. dorsalis*. The hybridization protocol for the gut and ovary was performed according to a previously described method[32]. Briefly, the gut and ovary were collected and immediately soaked in Carnoy's fixative for 12 h. After sample fixation, proteinase K (2 mg/mL) treatment for 20 min at 37 °C and HCl (0.2 mol/L) treatment for 15 min at room temperature were performed successively. Then, followed by dehydration in ethanol, the samples were incubated in buffer (20 mM Tris-HCl (pH 8.0), 0.9 M NaCl, 0.01% sodium dodecyl sulfate, 30% formamide) containing 50 nM CF-BD specific probe (5'-AATGGCGTACA-CAAAGAG-3') labeled with cy3 at the 5' end for 90 min. After incubation, the samples were washed with buffer (0.1 M NaCl, 20 mM Tris/HCl (pH 8.0), 5 mM ethylenediaminetetraacetic acid (pH 8.0), 0.01% SDS) and observed under an epifluorescence microscope (Axiophot, Carl Zeiss, Shinjuku-ku, Japan).

To further confirm CF-BD in rectum and ovary of mature females, rectums and ovaries of mature females were dissected and fixed in formalin fixation for 24 h. After soaking in graded alcohols and xylene, all samples were embedded in paraffin for section preparation. Samples were sliced into 4 μm each before pasting on the glass slide and then sent for FISH with the same probe (labeled with cy3 at the 5' end) used above. Moreover, nested PCR was applied to detect CF-BD in 19 ovaries of mature females according to the method of Guo et al., 2017[33]. Briefly, a 1149 bp region of gyrB gene of CF-BD was amplified by the specific outer primer gyrBP1-F (5'-CAGCCCACTCTGAACTGTAT-3') and gyrBP1-R (5'-TCAGGGCGTTTTCTTCGATA-3') under a temperature profile of 95 °C for 1 min, which was followed by 25 cycles of 95 °C for 30 s, 52 °C for 30 s, 72 °C for 90 s, and 72 °C for 5 min. Then, a 371 bp region of the gyrB gene of CF-BD was amplified by the specific inner primer gyrBP4-F (5'-ACGCTGGCTGAAGACTGCC-3') and gyrBP4-R (5'-TGGATAGCGAGACCACGACG-3') under a temperature profile of 95 °C for

2 min, which was followed by 35 cycles of 95 °C for 30 s, 57 °C for 30 s, 72 °C for 30 s, and 72 °C for 5 min.

**Influence of CF-BD on *B. dorsalis* ovary development.** To evaluate the effect of CF-BD on ovary development, newly emerged *B. dorsalis* females were injected with streptomycin and CF-BD suspension (both dilute in sterile water). Specifically, 10 μL 25% glycerol solution containing CF-BD was added into 100 mL Luria-Bertani (LB) liquid medium and culturing for 1 day by shaking (180 rpm) in 30 °C incubator. After culturing, CF-BD was collected by centrifuging (3000 rpm, 15 min) the medium in a 50 mL centrifuge tube. Then collected CF-BD was re-suspended with 5 mL sterile water. CF-BD concentration was measured on a hemocytometer and CF-BD concentrations used in the following assays were prepared by diluting the original concentration with sterile water. A 0.5 mm inside diameter capillary needle with 1 μL streptomycin or CF-BD suspension was used for injection. The injection operation was carried out on a microinjector (Eppendorf FemtoJet), and every female was injected in the abdomen near the ovipositor. The concentrations of streptomycin used were 20 mg/mL, 10 mg/mL and 5 mg/mL, respectively. And CF-BD suspension concentrations were $3 \times 10^7$ cfu/mL, $1.5 \times 10^7$ cfu/mL and $7.5 \times 10^6$ cfu/mL, respectively. For control, the female fly was injected with 1 μL sterile water in the abdomen near the ovipositor. Then the development level of the ovary was assessed by comparing the width and length of ovary between streptomycin (or CF-BD suspension) injection flies and control. For CF-BD injected flies, developmental facilitation was observed for ovaries 2 days before the flies reached sexual maturity (flies will reach sexual maturity after 7 days). For antibiotic injected flies, ovaries were dissected after 7 days.

**Oviposition assays.** The method reported in previous studies was followed for the oviposition experiments[17]. Briefly, a 2-choice apparatus was assembled in a cage made up of wood and wire gauze (length: width: height = 60 cm: 60 cm: 60 cm) with two petri dishes (diameter: 3 cm) at the bottom of the cage (Fig. 2a). All devices were sterilized before each experiment. Fresh fruits of guava (*Psidium guajava* Linn.) and mango (*Mangifera indica* L.) were sourced from the local market in Guangzhou, China. These fruits were sterilized on the surface with ethanol and ground into puree with a sterilized grinder, and puree (2 g) was added to the sterilized Petri dishes of the cages (one dish with puree containing 100 μL CF-BD (0.8*10^8 cfu/mL) in sterile water, and one dish with puree containing 100 μL sterile water). Then the prepared cages were divided into two groups for different assays. Group 1: At 0 h, 50 gravid females of *B. dorsalis* were placed in the cages and egg numbers in the petri dishes were recorded after 2 h. Group 2: At 4 h, 50 gravid females of *B. dorsalis* were placed in the cages and egg numbers in the petri dishes were recorded after 2 h.

To test the oviposition attraction of 3-HA, a 4-choice apparatus was assembled in a cage made up of wood and wire gauze (length: width: height = 60 cm: 60 cm: 60 cm) with four petri dishes (diameter: 3 cm) at the bottom of the cage. In the Petri dishes, 2 g puree, 2 g puree + 0.2 mg 3-HA, 2 g puree + 2 mg 3-HA and 2 g puree + 20 mg 3-HA were added. Then, the egg-laying behavior was observed[31].

To test the oviposition attraction of 3-HA to flies with genes knocked down, 20 females injected with dsRNA were placed into the above cage with two Petri dishes. In the Petri dishes, 2 g guava puree and 2 g guava puree + 20 mg 3-HA were added. Then, the egg-laying behavior was observed using the above method. Oviposition of normally reared females was performed as a control. The oviposition index was calculated using the following formula:

Oviposition index = $(O − C)/(O + C)$, where O is the number of eggs in the treatment and C is the number of eggs in the control.

**Volatile analysis.** The volatile compounds in guava and mango purees were analyzed by GC–MS according to the method described in a previous study[17]. Briefly, 2 g puree mixed with sterile water or CF-BD was added into a 20 ml bottle, and then a 100-μm polydimethylsiloxane (PDMS) SPME fiber (Supelco) was used to extract the headspace volatiles for 30 min. GC–MS was performed with an Agilent 7890B Series GC system coupled to a quadruple-type-mass-selective detector (Agilent 5977B; transfer line 250 °C, source 230 °C, ionization potential 70 eV). The 3-HA concentrations in puree mixed with sterile water and CF-BD were measured with the standard curve drawn by the authentic standards of 3-HA. And 3-HA concentration in puree mixed with sterile water and CF-BD was compared with a paired sample Student's *t*-test.

**Olfactometer bioassays.** An olfactometer consisting of a Y-shaped glass tube with a main arm (20 cm length*5 cm diameter) and two lateral arms (20 cm length, 5 cm diameter) was used. The lateral arms were connected to glass chambers (20 cm diameter, 45 cm height) in which the odor sources were placed. To ensure a supply of odor-free air, both arms of the olfactometer received charcoal-purified and humidified air at a rate of 1.3 L/min.

To test the attraction effect of puree supplemented with CF-BD or 3-HA for females, puree mixed with CF-BD was prepared and placed in one odor glass chamber. In the control odor glass chamber, puree mixed with sterile water was placed. After 4 h, gravid females were individually released at the base of the olfactometer and allowed 5 min to show a selective response. The response was recorded when a female moved >3 cm into one arm and stayed for >1 min. Females

that did not leave the base of the olfactometer were recorded as nonresponders. Only females that responded were included in the data analysis. Odor sources were randomly placed in one arm or the other at the beginning of the bioassay, and the experiment was repeated ten times. The system was washed with ethanol after every experiment. More than 100 females were selected for testing, and each female was used only once for each odor. A chi-square test was performed to compare the attraction difference between puree mixed with sterile water and CF-BD.

**Olfactory trap assays.** The attraction of purees supplemented with CF-BD to mature females was also tested. The test chamber was assembled with a plastic cylinder (120 × 30 cm) covered by a ventilated lid. The test chamber contained an odor-baited trap (2 g puree + 100 μL CF-BD ($0.8*10^8$ cfu/mL)) and a control trap (2 g puree + 100 μL sterile water). The traps were made of transparent plastic vials (20 × 6 cm) and were sealed with a yellow lid on which small entrances were present to let the flies in (Fig. 3a). After 0 h or 4 h of fermentation, 100 gravid females were released in the cage. The fly number in each trap bottle was recorded after 2 h. The number of flies was compared with a paired sample Student's t-test.

The attraction effect of puree supplemented with 3-HA on mature females was tested by placing four traps (2 g puree, 2 g puree + 0.2 mg 3-HA, 2 g puree + 2 mg 3-HA and 2 g puree + 20 mg 3-HA) in the test chamber. Then, the attraction effect was observed[31].

**Video observation of egg-laying behavior.** Egg-laying behavior was observed in a Petri dish. Briefly, guava puree was added to a centrifuge tube on which a hole was made. Then, one gravid female was placed into the petri dish, and the lid was closed. Above the petri dish, a camera was placed to record the behavior of the female before laying eggs.

**EAG analysis.** EAG analysis was performed to determine whether 3-HA could elicit electrogram responses in the ovipositors of gravid females and Obps knocked down gravid females. For EAG preparations, the ovipositor of a gravid female was cut off and mounted between two glass electrodes (one electrode connected with the ovipositor tip). The ovipositor tip was cut slightly to facilitate electrical contact. Dilution of 3-HA in ethanol (0.1, 1 and 10 mg/mL) was used as a stimulant. Ethanol was used as control. For each ovipositor, ethanol and 3-HA diluted in ethanol were used as stimulants. The signals from the ovipositors were analyzed with GC-EAD 2014 software (version 4.6, Syntech).

**Transcriptome sequencing and gene identification.** To identify the olfactory genes that contribute to *B. dorsalis* oviposition preference, the transcriptome sequencing results of the female ovipositors at different developmental times (0 day, 3 days, 6 days, 9 days and 12 days) were compared. For each time, 5 ovipositors were dissected for RNA extraction. In addition, five replicates were included for each time. In the next step, paired-end RNA-seq libraries were prepared by following Illumina's library construction protocol. The libraries were sequenced on an Illumina HiSeq2000 platform (Illumina, USA). FASTQ files of raw reads were produced and sorted by barcodes for further analysis. Prior to assembly, paired-end raw reads (uploaded to National Genomics Data Center, Accession number: PRJCA004790) from each cDNA library were processed to remove adapters, low-quality sequences ($Q < 20$), and reads contaminated with microbes. The clean reads were de novo assembled to produce contigs. An index of the reference genome of *B. dorsalis* was built, and paired-end clean reads were mapped to the reference genome using HISAT2. 2.4 with "-rna-strandness RF" and other parameters set as a default[68]. To evaluate transcript expression abundances, StringTie software was applied for calculation of the normalized gene expression value FPKM[69]. Then, gene differential expression analysis was performed with DESeq2 software[70]. Genes/transcripts with a false discovery rate (FDR) below 0.05 and absolute fold change ≥2 were considered differentially expressed genes/transcripts. According to the above results, we screened the differentially expressed olfactory genes.

**Trend analysis.** Gene expression pattern analysis was used to cluster genes with similar expression patterns for the ovipositor samples. To examine the expression pattern of the differentially expressed genes, the expression data of each sample were normalized and then clustered with Short Time-series Expression Miner software (STEM)[71]. The clustered profiles with $p < 0.05$ were considered significant profiles.

**Expression validation of olfactory genes.** qRT-PCR analysis was used to validate olfactory gene expression in ovipositors at different developmental times. Antennae of 12-day-old female were used as controls. Total RNA from the antennae of 12-day-old females and ovipositors at 0 d, 3 d, 6 d, 9 d and 12 d was extracted. Then, cDNA was synthesized with a One-Step gDNA Removal and cDNA Synthesis SuperMix Kit (TransGen Biotech, Beijing, China) using the extracted RNA. Then, a PerfectStarTM Green qPCR SuperMix Kit (TransGen Biotech, Beijing, China) was used to perform quantitative real-time PCR to compare the gene expression levels. Gene-specific primers (Supplementary Table 1) were designed on NCBI with

primer blast. The *α-tubulin* and *actin* genes were used as reference genes[72]. The PCR procedure was set according to the manufacturers' instruction. Three biological duplicates and 3 technical duplicates were performed.

**The ability of the Obps to bind 3-HA.** Recombinant proteins Obp56d and Obp56d-2 were obtained by in vitro expression in *Escherichia coli* Rosetta (DE3) cells mainly according to the previous study[73]. Briefly, PCR primers (Supplementary Table 2) with restriction sites were designed according to the CDS sequences of *Obp56d* and *Obp56d-2*. Then PCR products were purified and connected to the peT-32A (+) Prokaryotic expression vector. The vector was then transformed into *Escherichia coli* Rosetta (DE3) for expression. The positive clones were screened by kanamycin resistance and sequenced to confirm the correct sequence. The verified clones were cultured in the kanamycin resistance LB medium at 37 °C for 16 h. Then 100 ml bacterial solution was inoculated in the LB medium with the final concentration of 0.1 mM IPTG for 8 h at 18 °C. Then bacteria were collected by centrifugation at 8000 rpm and then re-suspended in lysis buffer (80 mM Tris-HCl, 200 mM NaCl, 1 mM EDTA, 4% glycerol, pH 7.2, 0.5 mM PMSF). Sonication (3 s, five passes) was conducted to break the bacteria cells. The recombinant proteins in the supernatant were collected by centrifugation. Then the proteins were purified by two rounds of anion-exchange chromatography and concentrated by the ultrafiltration cube. Both purified recombinant protein was detected using SDS-PAGE.

The combination fluorescence experiment of Obp with N-phenyl-1-naphthylamine (1-NPN) was performed on a Microplate Reader (Thermo Scientific Varioskan LUX) referring the previous study in locust[74]. The excitation wavelength was set as 337 nm, and the emission wavelength was 380-520 nm. Obp (2 μM dissolved in 50 mM Tris-HCl, PH = 7.4) and 1-NPN (2μM–16μM dissolved in chromatographic methanol) were mixed and the strongest fluorescence values was recorded. The dissociation constant value of 1-NPN combined with Obp was calculated using one site-specific binding method in GraphPad 8.0. Then the dissociation constants of the Obp56d and Obp56d-2 were generated. For competitive binding assays, 3-HA (0, 4, 8, 12, 16 μM dissolved in chromatographic methanol) was used as competitor to bind the Obp in the Obp/1-NPN complex.

**RNA interference of the olfactory genes.** Double-stranded RNA (dsRNA) primers (Supplementary Table 2) tailed with the T7 promoter sequence were designed using the CDSs of *Obp56d, Obp56d-2, OR7a-2* and *IR25a* as templates. A MEGAscript RNAi Kit (Thermo Fisher Scientific, United States) was used to synthesize and purify dsRNA according to the manufacturer's instructions. The *GFP* gene was set as the RNAi negative control. To knockdown the target gene in females, 0.5 μL (500 ng/μl) dsRNA was injected into the abdomen of an 11-day-old female. Flies injected with ds*GFP* were prepared as a negative control. After 24 h, the knockdown efficiency of the genes was checked with qRT-PCR following the method used for validating the expression of the olfactory genes above. Then, the oviposition preference for 3-HA was tested in flies in which the olfactory gene was silenced.

**Statistics and reproducibility.** In box-and-whisker plots, where the boxes encompass the first to the third quartiles, inside the box the horizontal line shows the median, and the whiskers are the maximum and minimum observation. In bar plots, the data is presented as mean ± SEM. ANOVA followed by Tukey's test, Wilcoxon matched-pairs signed rank test, paired sample student t-test, Kendall nonparametric test, chi-square test and Kruskal–Wallis test followed by Dunn's test were used to compare difference between experimental treatments. In ANOVA, Kruskal–Wallis test is used instead for the data don't conform to ANOVA assumptions. Prism 8 (GraphPad, San Diego, CA) was used to do statistical analysis and generate the figures. $P \leq 0.05$ were considered as statistically significant. Sample sizes are in all cases stated in the manuscript.

**Reporting summary.** Further information on research design is available in the Nature Research Reporting Summary linked to this article.

## Data availability

All data needed to evaluate the conclusions in the paper are present in the paper and/or the Supplementary Materials. Raw data, replicates and statistical analysis methods in the figures are listed in Supplementary Data 2. Supplementary Movie 1 was uploaded in Figshare (https://figshare.com/articles/media/Gut_bacteria_induced_oviposition_preference_through_ovipositor_recognition/20575248). The transcriptome data of ovipositor have been uploaded to National Genomics Data Center with accession number being PRJCA004790. Accession number of *GFP, Obp56d, Obp56d-2, Obp19d, OR7a-2* and *IR25a* in GenBank were AHE38523, XM_011198997.3, XM_011198998.2, XM_011213044.3, KP743713.1 and XM_011209493.3, respectively.

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

## Acknowledgements
We are grateful to the national natural science foundation of China (No. 3212200346).

## Author contributions
M. H., H. C., X. Y. and Y. G. conducted the experiments; D. C. designed the experiments; D. C. and M. H. wrote the paper. D. C. and Y. L. revised the manuscript.

## Competing interests
The authors declare no competing interests.
