## [Peer Review File · Communications Biology]

Reviewers' comments:

Reviewer #1 (Remarks to the Author):

In their manuscript "gut bacteria induced oviposition preference through ovipositor recognition" the authors describe the effect of gut associated *Citrobacter* sp. (CF-BD) on ovary growth/maturation in *Bactrocera dorsalis* females and their release of 3-hexenyl acetate on food which acts as a oviposition cue, perceived via olfactory receptors present on female ovipositors. The manuscript is overall based on sound evidence with interesting, although not completely novel findings (bacterial oviposition cues, ovipositor localized olfactory reception). Nevertheless, the combined results will be interesting for the both relatively young fields of research and beyond. However, I have a few concerns regarding their statistical procedures, details of their experiments and conclusions that should be addressed before considering publication of the manuscript.

1. Localization of CF-BD in ovaries

How do you explain discrepancies on CF-BD localization in ovaries in this study versus Guo et al who did not detect them in ovaries of *B. dorsalis*, neither by FISH, nor PCR. As the FISH signal is not highly resolved, and no control experiments are presented in this study I am not entirely convinced this is indeed a specific signal, but could also be background or autofluorescence, e.g. caused by fixation artefacts or non-specific binding of the probe; I would suggest to complement the FISH experiment with controls e.g. use anti-sense or off-target probe, or higher resolution images verifying signal is localized from bacterial cells, e.g. HR-confocal or widefield fluorescence of sections, or standard histological sections and staining to verify presence of bacteria throughout the ovaries. Alternatively, but with lower resolution PCR of dissected tissues.

How frequent is the invasion of ovaries by CF-BD across specimen?

2. Effect of CF-BD on ovary development

In line 65 you conclude that the growth differences of ovaries are caused by CF-BD in ovaries. I don't see evidence that the bacteria need to be specifically localized in the ovaries. It could also be a general nutritional effect. Esp. as the manuscript doesn't contain information about frequency of ovary infection (see also 1.)

3. Statistical tests

The datasets of figures 2B, 2E, 3B, 3C contain dependent data, basically always proportional data as females could choose to oviposit in one of multiple presented food sources. I think t-tests are not appropriate for this kind of datasets.

For several other experiments the statistical tests are not described, also the tests for assumptions of the following tests are missing (data distribution, homogeneity of variances): Figures 1F, 1D, 3C, 4B, 4D-H, 5. Test-statistics should be also described in more detail, e.g. as requested by the Reporting summary: Easily identifiable number of replicates, primary test values (t, Chi2...), precise p-values, what the horizontal lines in graphs represent: means or medians.

Is the raw data of all experiments (beyond sequence data) deposited in any archive? The Data availability statement is not clear in that regard.

The tests for differences between compounds identified in the SPME-GC-MS experiments should be corrected for false discovery rates of multiple testing.

4. Implications of aggregation induction by symbionts

Regarding statements in your concluding paragraph, a consequence of the attraction of CF-BD and their transmission with deposited eggs is aggregation of eggs from multiple conspecifics. This could have both beneficial and detrimental effects, e.g. intraspecific competition or synergies in regard of diet or defense, with evolutionary feedback on the transmission / attraction. Does literature on *B. dorsalis* reproduction and development indicate any of these effects?

Minor comments:

Literature on symbiont mediated manipulation of insect reproduction / behavior is well reviewed and should be cited in the introduction; also specific literature on attraction / aggregation pheromone

producing bacteria/yeast/fungi in flies and beetles, possibly also by pathogens should be integrated in the discussion.

e.g.

Becher, P.G., Flick, G., Rozpędowska, E., Schmidt, A., Hagman, A., Lebreton, S., Larsson, M.C., Hansson, B.S., Piškur, J., Witzgall, P. and Bengtsson, M. (2012), Yeast, not fruit volatiles mediate *Drosophila melanogaster* attraction, oviposition and development. *Funct Ecol*, 26: 822-828. <https://doi.org/10.1111/j.1365-2435.2012.02006.x>

Lam, K. et al. Proliferating bacterial symbionts on house fly eggs affect oviposition behaviour of adult flies. *Anim. Behav.* 74, 81–92 (2007).

But also bark-beetle symbiont mediated aggregation.

L41-43: consider changing "social" to "gregarious" the two cited references are not

L73: change "females would prefer lay eggs" to "females prefer to lay eggs"

L102-103: remove one of the verbs

L215-216: please add amount of sugar, yeast extract and water volume.

Reviewer #2 (Remarks to the Author):

The authors are shedding light on the previously identified relationship of the oriental fruit fly, *Bactrocera dorsalis*, with its gut symbiont, *Citrobacter* sp. The hypothesis they present and test is that *Citrobacter* sp. is actively attracting *B. dorsalis* to lay eggs, through the recognition of 3-hexenyl acetate by olfactory gene products of the ovipositor and this leads to a better spread of bacteria in their host, what the authors consider as an efficient and novel route of gut bacteria transmission. To address this, the authors performed a series of well-executed experiments involving multidisciplinary approaches, which, step-by-step, built their argument. This can be considered a novel finding, at least for fruit flies.

The manuscript is in general well written. The introduction is clear, recognizes previous work in the field, and poses the question the authors are addressing. Experiments are well-performed and clearly explained apart from some concerns regarding experimental procedures and conclusions deduced can be found below. The figures are interesting and informative, needing some minor adjustments (see below).

1. L58-69: CF is clearly present in the ovaries. CF suspension is clearly increased when you use a CF suspension. However, how do the authors verify that streptomycin affects only CF and no other symbionts? A decrease in the development could be attributed to the reduction of other symbionts, apart from CF. On the same topic, although the increase in the CF is correlated with increased ovaries development, is it possible that this is not a CF-specific phenomenon but more like a universal symbiont-increase effect?
2. L70-82: again, is the specific CF strain the only symbiont having these properties? Is 3-HA specific for CF or it can be produced by other symbionts? This applies to all downstream results. To make it clearer: the authors document the properties of the CF strains, however, can they support that these properties are restricted only to the specific symbiont and not others?
3. L146-147: what means 'a new role evolved'? This implies that this is something novel and not present in other species.
4. L204-206: starting two consecutive sentences with 'on the other hand' is rather confusing.
5. L229-239: details for the cultivation of CF are needed (media and conditions for cultivation, how cfu/ml was documented).
6. L236-239: How was the ovaries' development assessed? How it was different between CF and antibiotic treatment? Does the injection itself have a cost for survival or ovary development?

7. Line 250: the authors say they used 'gravid females'. Was this verified somehow?
8. L386: the qRT-PCR was performed as above? Details are needed.
9. Materials and Methods: a clear description of the statistical analysis is missing in this part (only evident in the Figures). Different tests have been used but it is not specified how they were performed or why they are the appropriate tests (for ANOVA, for example, whether the data conform to ANOVA assumptions). A supplement giving the raw data and complete statistical analysis would probably be beneficial for this submission.
10. Figures are nice and informative. However, they are really 'dense' and therefore not clear in some aspects (see blue spots and error bars in the graphs)

Reviewer #3 (Remarks to the Author):

This paper describes a series of experiments on a bacterial symbiont (CF-BD) of the oriental fruit fly. This research group has made several exciting and novel advances in our understanding of fruit fly gut symbionts. Previous work has shown that that CF-BD can be vertically transmitted from mothers to offspring, and horizontally between larvae within fruit (Guo et al., 2017). Additionally, this bacterium is active in conferring resistance to an insecticide (Cheng et al., 2017). Concurrently, other egg-surface bacteria such as *Klebsiella* are indirectly associated with oviposition aversion in *Bactrocera dorsalis* (Li et al 2020).

Here, in a series of elegant experiments, He et al., show that CF-BD can move from the rectum to the ovaries of *Bactrocera dorsalis* females, where they are associated with ovarian development. Gravid females were attracted to fruit substrates containing active CF-BD. These substrates produced 3-hexenyl acetate (3-HA), which independently was shown to attract gravid females. Olfactometer assays showed that the attraction to CF-BD and 3-HA is not mediated by the antennae or over long distances. Notably, evidence is provided that sensilla on the ovipositor respond to 3-HA and express genes for odorant binding proteins. This is a highly novel finding (smelling through the ovipositor!). Sophisticated knockdown experiments with EAG pinpoint the identity of 2 ObP's in the ovipositors response to volatiles. Again this result is highly novel, and presented with rigor.

I have no major comments regarding the experiments or their analysis, although some more details and a few clarifications may be helpful and are suggested (see attached ms. file).

The only problem I have with this manuscript is that the authors fail to place the findings in context, both from a broad perspective, and a specific one.

The broad context is of Behavioral Microbiomics, and a list of some recent reviews and relevant studies follows.

The specific context should be one whereby the results are interpreted in an ecological perspective that is relevant to the fly, and to the accumulated knowledge about its oviposition decisions and bacterial interactions. This is missing and should be addressed in the revision.

Starting from the introduction, circular inferences are made regarding the "motivation" of CF-BD in attracting the fly. For example: "We infer that the reproduced CF-BD may attract females to lay eggs in CF-BD-infected fruits to ensure that CF-BD can be effectively transmitted in the population of *B. dorsalis*". or: "By attracting the females to lay eggs, CF-BD can be effectively transmitted in the host populations, and the host populations can benefit from CF-BD"; or "To ensure that the larvae can acquire CF-BD, females will lay eggs in CF-BD-colonized fruits". This is a circular argument, that fails to address how CF-BD gets to the fruit in the first place. *Bactrocera* do not oviposit preferentially in rotting fruit, and Li et al have shown that females use bacteria (*Providencia* sp. and *Klebsiella* sp) to deter other females from ovipositing. Ovipositing in a fruit that is occupied by other larvae may be maladaptive for females, and indeed they prefer intact fruit over rotting fruit (eg., Diaz-Fleischer et al.; Silva & Clarke). Nevertheless, this study finds a strong biological link between the bacteria and the fly. This apparent contradiction must be addressed in the revision.

Congratulations to all involved.

Some references:

- Akami, M., Andongma, A. A., Chen, Z., Nan, J., Khaeso, K., Jurkevitch, E., . . . Yuval, B. (2019). Intestinal bacteria modulate the foraging behavior of the oriental fruit fly *Bactrocera dorsalis* (Diptera: Tephritidae). *Plos One*, 14(1). doi: 10.1371/journal.pone.0210109
- Diaz-Fleischer, F., Papaj, D. R., Prokopy, R. J., Norrbom, A. L., & Aluja, M. (2000). Evolution of fruit fly oviposition behavior. In M. Aluja & A. L. Norrbom (Eds.), *Fruit Flies (Tephritidae): Phylogeny and Evolution of Behavior* (pp. 811-841.). Boca Raton: CRC.
- Shu, R., Hahn, D. A., Jurkevitch, E., Liburd, O. E., Yuval, B., & Wong, A. C.-N. (2021). Sex-Dependent Effects of the Microbiome on Foraging and Locomotion in *Drosophila suzukii*. [Original Research]. *Frontiers in Microbiology*, 12(1094). doi: 10.3389/fmicb.2021.656406
- Silva, R., & Clarke, A. R. (2021). Aversive responses of Queensland fruit flies towards larval-infested fruits are modified by fruit quality and prior experience. *Journal of Insect Physiology*, 131. doi: 10.1016/j.jinsphys.2021.104231
- Sivakala, K. K., Jose, P. A., Shamir, M., Wong, A. C. N., Jurkevitch, E., & Yuval, B. (2022). Foraging behaviour of medfly larvae is affected by maternally transmitted and environmental bacteria. *Animal Behaviour*, 183, 169-176. doi: 10.1016/j.anbehav.2021.10.014
- Wong, A. C. N., Holmes, A., Ponton, F., Lihoreau, M., Wilson, K., Raubenheimer, D., & Simpson, S. J. (2015). Behavioral Microbiomics: A Multi-Dimensional Approach to Microbial Influence on Behavior. *Frontiers in Microbiology*, 6.
- Wong, A. C. N., Wang, Q. P., Morimoto, J., Senior, A. M., Lihoreau, M., Neely, G. G., . . . Ponton, F. (2017). Gut Microbiota Modifies Olfactory-Guided Microbial Preferences and Foraging Decisions in *Drosophila*. *Current Biology*, 27(15), 2397-+.

Reviewer #1 (Remarks to the Author):

In their manuscript “gut bacteria induced oviposition preference through ovipositor recognition” the authors describe the effect of gut associated *Citrobacter* sp. (CF-BD) on ovary growth/maturation in *Bactrocera dorsalis* females and their release of 3-hexenyl acetate on food which acts as a oviposition cue, perceived via olfactory receptors present on female ovipositors. The manuscript is overall based on sound evidence with interesting, although not completely novel findings (bacterial oviposition cues, ovipositor localized olfactory reception). Nevertheless, the combined results will be interesting for the both relatively young fields of research and beyond. However, I have a few concerns regarding their statistical procedures, details of their experiments and conclusions that should be addressed before considering publication of the manuscript.

Response to reviewer: Thank you very much for your comments on our manuscript. And we have studied your comments carefully and made some revisions to address the points raised. We hope the revision will meet with approval.

1. Localization of CF-BD in ovaries

How do you explain discrepancies on CF-BD localization in ovaries in this study versus Guo et al who did not detect them in ovaries of *B. dorsalis*, neither by FISH, nor PCR. As the FISH signal is not highly resolved, and no control experiments are presented in this study I am not entirely convinced this is indeed a specific signal, but could also be background or autofluorescence, e.g. caused by fixation artefacts or non-specific binding of the probe; I would suggest to complement the FISH experiment with controls e.g. use anti-sense or off-target probe, or higher resolution images verifying signal is localized from bacterial cells, e.g. HR-confocal or widefield fluorescence of sections, or standard histological sections and staining to verify presence of bacteria throughout the ovaries. Alternatively, but with lower resolution PCR of dissected tissues. How frequent is the invasion of ovaries by CF-BD across specimen?

Response to reviewer: Thanks for your comments. Actually, our results do not contradict the conclusions of Guo et al. The reason that Guo et al did not detect CF-BD in ovaries of *B. dorsalis* is that the ovaries they dissected were from the immature females (newly emerged females). As you can see in Figure 1A, no CF-BD signal can be detected in immature ovaries. Besides, Guo et al had found that 100% eggs carried CF-BD. One possible reason is that the eggs get these bacteria from the ovaries. To further address your question, we have done the standard histological sections and staining to verify presence of bacteria in rectums and ovaries. The results also indicated that CF-BD can be detected in rectums and ovaries. And nested PCR also shown that all ovaries (15) of mature females contain CF-BD. See line 61-69. Thanks for your valuable comments and we hope that the correction will meet with approval.

2. Effect of CF-BD on ovary development

In line 65 you conclude that the growth differences of ovaries are caused by CF-BD in ovaries. I don't see evidence that the bacteria need to be specifically localized in the ovaries. It could also be a general nutritional effect. Esp. as the manuscript doesn't contain information about frequency of ovary infection (see also 1.)

Response to reviewer: Thanks for your valuable comments. First of all, it could not be a general nutritional effect, since we have diluted the bacteria in sterile water before injection. We have

mentioned this in the “material and method” part. Second, we have added the standard histological sections and staining assays to detect CF-BD in ovaries of mature females and the results indicated that ovaries of mature females carried CF-BD. Moreover, nested PCR results also indicated that 100% ovaries of mature females carried CF-BD. We agree with your opinion that we can’t make conclusion that ovary development was affected by CF-BD in ovary. To address such issue, we have rephrased the statement in the manuscript by saying that CF-BD or streptomycin injection could affect ovary development. Please see line 61-69. Thank you very much for your constructive comments. We hope that revision and the explanation will meet with approval.

3. Statistical tests

The datasets of figures 2B, 2E, 3B, 3C contain dependent data, basically always proportional data as females could choose to oviposit in one of multiple presented food sources. I think t-tests are not appropriate for this kind of datasets.

Response to reviewer: We are very sorry for the mistakes we have made in statistical analysis. We have re-analysis the data. For Figure 2B, Wilcoxon matched-pairs signed rank test was used; For Figure 2E, the Kendall nonparametric test was used; For Figure 3B, Wilcoxon matched-pairs signed rank test was used; For Figure 3C, the Kendall nonparametric test was used. We have added the statements for statistical tests in the figure legends and added a Dataset 2 to include the raw data and statistical analysis results. Please see figure legends and Dataset 2. Thanks for your valuable comments and we hope that the correction will meet with approval.

For several other experiments the statistical tests are not described, also the tests for assumptions of the following tests are missing (data distribution, homogeneity of variances): Figures 1F, 1D, 3C, 4B, 4D-H, 5. Test-statistics should be also described in more detail, e.g. as requested by the Reporting summary: Easily identifiable number of replicates, primiar test values (t, Chi2...), precise p-values, what the horizontal lines in graphs represent: means or medians.

Is the raw data of all experiments (beyond sequence data) deposited in any archive? The Data availability statement is not clear in that regard.

The tests for differences between compounds identified in the SPME-GC-MS experiments should be corrected for false discovery rates of multiple testing.

Response to reviewer: Thank you very much for your constructive comments. To address the issues you have mentioned we have added a Dataset 2 to include the raw data and the statistical tests results of each figure. For compounds identified in GC-MS, we have made the paired sample t test in guava (there is no need for statistical test in mango, since no 3-HA was detected in control mango puree), see Figure 2E. Thanks for your valuable comments and we hope that the correction will meet with approval.

4. Implications of aggregation induction by symbionts

Regarding statements in your concluding paragraph, a consequence of the attraction of CF-BD and their transmission with deposited eggs is aggregation of eggs from multiple conspecifics. This could have both beneficial and detrimental effects, e.g. intraspecific competition or synergies in regard of diet or defense, with evolutionary feedback on the transmission / attraction. Does literature on *B. dorsalis* reproduction and development indicate any of these effects?

Response to reviewer: Thanks for your comments. We agree with you that aggregation of eggs could have both beneficial and detrimental effects. Literatures have also indicated such effects in *B. dorsalis*. Previous study has shown that gravid female of *B. dorsalis* will usually lay dozens of eggs in one fruit. However, our recent study indicated that too many maggots in fruit will repel oviposition of the gravid females. Thus, it's an issue of how the flies evaluate how many eggs they should lay in a fruit. We have discussed this in discussion. See line 153-166. Thanks for your valuable comments and we hope that the correction will meet with approval.

Minor comments:

Literature on symbiont mediated manipulation of insect reproduction / behavior is well reviewed and should be cited in the introduction; also specific literature on attraction / aggregation pheromone producing bacteria/yeast/fungi in flies and beetles, possibly also by pathogens should be integrated in the discussion.

e.g.

Becher, P.G., Flick, G., Rozpędowska, E., Schmidt, A., Hagman, A., Lebreton, S., Larsson, M.C., Hansson, B.S., Piškur, J., Witzgall, P. and Bengtsson, M. (2012), Yeast, not fruit volatiles mediate *Drosophila melanogaster* attraction, oviposition and development. *Funct Ecol*, 26: 822-828. <https://doi.org/10.1111/j.1365-2435.2012.02006.x>

Lam, K. et al. Proliferating bacterial symbionts on house fly eggs affect oviposition behaviour of adult flies. *Anim. Behav.* 74, 81–92 (2007).

But also bark-beetle symbiont mediated aggregation.

Response to reviewer: Thanks for you valuable suggestions. We have added the information in “introduction” and “discussion” part. See line 20-21, line 32-36 and line 182-185.

L41-43: consider changing “social” to “gregarious” the two cited references are not

Response to reviewer: done.

L73: change “females would prefer lay eggs” to “females prefer to lay eggs”

Response to reviewer: done.

L102-103: remove one of the verbs

Response to reviewer: done.

L215-216: please add amount of sugar, yeast extract and water volume.

Response to reviewer: done.

Reviewer #2 (Remarks to the Author):

The authors are shedding light on the previously identified relationship of the oriental fruit fly, *Bactrocera dorsalis*, with its gut symbiont, *Citrobacter* sp. The hypothesis they present and test is that *Citrobacter* sp. is actively attracting *B. dorsalis* to lay eggs, through the recognition of 3-hexenyl acetate by olfactory gene products of the ovipositor and this leads to a better spread of

bacteria in their host, what the authors consider as an efficient and novel route of gut bacteria transmission. To address this, the authors performed a series of well-executed experiments involving multidisciplinary approaches, which, step-by-step, built their argument. This can be considered a novel finding, at least for fruit flies.

Response to reviewer: Thank you very much for your positive comments.

The manuscript is in general well written. The introduction is clear, recognizes previous work in the field, and poses the question the authors are addressing. Experiments are well-performed and clearly explained apart from some concerns regarding experimental procedures and conclusions deducted can be found below. The figures are interesting and informative, needing some minor adjustments (see below).

Response to reviewer: Thank you very much for your constructive comments. We have addressed the issues you have mentioned point by point (please see below). Thanks for your valuable comments and we hope that the correction will meet with approval.

1. L58-69: CF is clearly present in the ovaries. CF suspension is clearly increased when you use a CF suspension. However, how do the authors verify that streptomycin affects only CF and no other symbionts? A decrease in the development could be attributed to the reduction of other symbionts, apart from CF. On the same topic, although the increase in the CF is correlated with increased ovaries development, is it possible that this is not a CF-specific phenomenon but more like a universal symbiont-increase effect?

Response to reviewer: We agree with you opinion. We have realized that we can't rule out that antibiotics don't affect other bacteria. Thus, we assessed ovary development in both elevated and decreased bacterial conditions. From these data, we can at least be sure that ovary development is associated with CF. However, it is almost impossible to exclude the effect of other symbionts. To make the statement more exact, we have rephrased the conclusion of this part. See line 61-69. And we have discussed this in discussion. Please see line 185-190. Thanks for your valuable comments and we hope that the correction will meet with approval.

2. L70-82: again, is the specific CF strain the only symbiont having these properties? Is 3-HA specific for CF or it can be produced by other symbionts? This applies to all downstream results. To make it clearer: the authors document the properties of the CF strains, however, can they support that these properties are restricted only to the specific symbiont and not others?

Response to reviewer: Thank you very much for you important question. This is vital to the conclusions in our manuscript. As you know there are numerous vertical transmitted symbionts in *B. dorsalis*, it is almost impossible for us to test if other bacteria have these properties. However, we have done the related study for the dominant vertical transmitted bacteria in our previous study (please see Li et al., 2020 Current Biology) and we have found that the tested dominant bacteria did not have these properties. Most importantly, our previous studies have shown that there was no 3-HA production if all the bacteria get into the fruit via oviposition. In this regard, it should be considered that only CF has these properties. We have discussed this in discussion. Please see line 185-190. Thank you again for you important question and we hope that the explanation will meet with approval.

3. L146-147: what means 'a new role evolved'? This implies that this is something novel and not present in other species.

Response to reviewer: Sorry for overstating our results. We have rephrased the statement.

4. L204-206: starting two consecutive sentences with 'on the other hand' is rather confusing.

Response to reviewer: Sorry for our writing problem. We have rephrased these sentences.

5. L229-239: details for the cultivation of CF are needed (media and conditions for cultivation, how cfu/ml was documented).

Response to reviewer: We are sorry for missing the information. We have added the information in the revised manuscript. See line 258-264.

6. L236-239: How was the ovaries' development assessed? How it was different between CF and antibiotic treatment? Does the injection itself have a cost for survival or ovary development?

Response to reviewer: We are sorry for missing the detail information. The development level of the ovary was assessed by comparing the width and length of ovary between streptomycin (or CF-BD suspension) injection flies and control. For control, the female fly was injected with 1 μ L sterile water in the abdomen near the ovipositor. By setting control, we can exclude the injection effect on survival or ovary development. See line 267-272. Thank you very much for your valuable questions. We hope that the revision will meet with approval.

7. Line 250: the authors say they used 'gravid females'. Was this verified somehow?

Response to reviewer: Yes. For *B. dorsalis*, the female will start laying eggs once mated and the female will start mating 7 days after emergence. To make sure all females were gravid females, females were selected 10 day after emergence. We have added such information in the manuscript. Please see line 235-237.

8. L386: the qRT-PCR was performed as above? Details are needed.

Response to reviewer: Yes, qRT-PCR was performed as above method. We have mentioned this in the revised manuscript. See line 423-424.

9. Materials and Methods: a clear description of the statistical analysis is missing in this part (only evident in the Figures). Different tests have been used but it is not specified how they were performed or why they are the appropriate tests (for ANOVA, for example, whether the data conform to ANOVA assumptions). A supplement giving the raw data and complete statistical analysis would probably be beneficial for this submission.

Response to reviewer: Thank you very much for your constructive comments. To address the issues you have mentioned we have added a Dataset 2 to include the raw data and the statistical tests results of each figure. In ANOVA, Kruskal-Wallis test is used instead for the data don't conform to ANOVA assumptions. We have mentioned this in the "Statistics and Reproducibility" part. Please see line 427-432.

10. Figures are nice and informative. However, they are really ‘dense’ and therefore not clear in some aspects (see blue spots and error bars in the graphs)

Response to reviewer: We are sorry for these issues. We have made revisions to all figures to make them clearer. According to the formatting guidelines of Communications Biology, we have converted all bar graphs to box-and-whisker to show data distribution.

Reviewer #3 (Remarks to the Author):

This paper describes a series of experiments on a bacterial symbiont (CF-BD) of the oriental fruit fly. This research group has made several exciting and novel advances in our understanding of fruit fly gut symbionts. Previous work has shown that that CF-BD can be vertically transmitted from mothers to offspring, and horizontally between larvae within fruit (Guo et al., 2017). Additionally, this bacterium is active in conferring resistance to an insecticide (Cheng et al., 2017). Concurrently, other egg-surface bacteria such as *Klebsiella* are indirectly associated with oviposition aversion in *Bactrocera dorsalis* (Li et al 2020).

Here, in a series of elegant experiments, He et al., show that CF-BD can move from the rectum to the ovaries of *Bactrocera dorsalis* females, where they are associated with ovarian development. Gravid females were attracted to fruit substrates containing active CF-BD. These substrates produced 3-hexenyl acetate (3-HA), which independently was shown to attract gravid females. Olfactometer assays showed that the attraction to CF-BD and 3-HA is not mediated by the antennae or over long distances. Notably, evidence is provided that sensilla on the ovipositor respond to 3-HA and express genes for odorant binding proteins. This is a highly novel finding (smelling through the ovipositor!). Sophisticated knockdown experiments with EAG pinpoint the identity of 2 ObP’s in the ovipositors response to volatiles. Again this result is highly novel, and presented with rigor.

Response to reviewer: Thank you very much for you positive comments.

I have no major comments regarding the experiments or their analysis, although some more details and a few clarifications may be helpful and are suggested (see attached ms. file).

Response to reviewer: Thank you again for your positive comments. We have addressed the issues you have mentioned. We hope that the revision will meet with approval.

But on line 71 you mention this was done in ref#23

Response to reviewer: Sorry for the incorrect statement. We have deleted such statement.

Mention presence in rectum here (Figure 1A), indicating that CF-BD can be transmitted from rectum into ovary???

Response to reviewer: Sorry for the inappropriate statement. We have rephrased the statement. We have realized that our data can only indicate that CF-BD can enter into the ovary at the mature stage. And we have added histological sections and staining results and nest PCR results to support such conclusion. Please see line 61-69. We hope that the revision will meet with approval.

Where does CFBD come from? Does it persist in pupae?

Response to reviewer: Thanks for your comments. The origination of CFBD is rather important.

And we're actually also interested in where CF-BD comes from. We have some clues about the origin of CF-BD from one of our previous studies (Zhao et al., 2017, *Frontiers in Microbiology*). It is likely that the CF-BD in the female is originated from the pupae.

What about larval competition? Deterrence by *Klebsiella*?

Response to reviewer: We have discussed this in “discussion” part. Female insects will take many aspects into account to give their offspring the best chances of survival, and select a unique oviposition site for maximizing the survival, growth, and reproductive potential of the offspring. Previous study has shown that gravid female of *B. dorsalis* will usually lay dozens of eggs in one fruit. However, our recent study has indicated that egg-surface bacteria induced volatile in fruit will repel oviposition of the gravid females. Thus, aggregation of eggs in a fruit may depend on the competition result of the egg-surface bacteria. Once the eggs were laid into the fruit, the egg-surface *Citrobacter* will be reproduced quickly and attract oviposition of *B. dorsalis*. However, other egg-surface bacteria (such as *Klebsiella*) will be reproduced at the stage that the maggots emerged and repel oviposition. See line 153-166. We hope that the explanation will meet with approval.

To ensure that the larvae can acquire CF-BD, females will lay eggs in CF-BD-colonized fruits.[Circular. where does it come from? Other females? What about competition?]

Response to reviewer: Thanks for your comments. Such questions are critical important. And we have answered in the above responses. We hope that the explanation will meet with approval.

Once *Citrobacter* is fed on by *B. dorsalis*, *Citrobacter* can help to digest fruit substances [this can happen in the fruit, not necessarily in the larval gut].

Response to reviewer: Thanks for your important comments. Here, we want to say “Once *Citrobacter* is fed by *B. dorsalis*, *Citrobacter* can help to digest fruit substances and make fruit substances being easily absorbed by larvae.” And we have revised the statement. We hope that the revision will meet with approval.

Do females prefer to oviposit in rotting fruit? With competing larvae?

Response to reviewer: Thanks for your comments. Such questions are critical important. And we have answered in the above responses and made revision in the manuscript.

The only problem I have with this manuscript is that the authors fail to place the findings in context, both from a broad perspective, and a specific one.

The broad context is of Behavioral Microbiomics, and a list of some recent reviews and relevant studies follows.

Response to reviewer: Thanks for your valuable comments. We have added the broad context for Behavioral Microbiomics in introduction and cited the relevant studies. See line 34-38. We hope that the revision will meet with approval.

The specific context should be one whereby the results are interpreted in an ecological perspective that is relevant to the fly, and to the accumulated knowledge about its oviposition decisions and bacterial interactions. This is missing and should be addressed in the revision.

Response to reviewer: Thanks for your valuable comments. We have tried our best to address the issues in discussion. See line 153-166. We hope that the revision will meet with approval.

Starting from the introduction, circular inferences are made regarding the “motivation” of CF-BD in attracting the fly. For example: “We infer that the reproduced CF-BD may attract females to lay eggs in CF-BD-infected fruits to ensure that CF-BD can be effectively transmitted in the population of *B. dorsalis*”. or: “By attracting the females to lay eggs, CF-BD can be effectively transmitted in the host populations, and the host populations can benefit from CF-BD”; or “To ensure that the larvae can acquire CF-BD, females will lay eggs in CF-BD-colonized fruits”. This is a circular argument, that fails to address how CF-BD gets to the fruit in the first place. *Bactrocera* do not oviposit preferentially in rotting fruit, and Li et al have shown that females use bacteria (*Providencia* sp. and *Klebsiella* sp) to deter other females from ovipositing. Ovipositing in a fruit that is occupied by other larvae may be maladaptive for females, and indeed they prefer intact fruit over rotting fruit (eg., Diaz-Fleischer et al.; Silva & Clarke). Nevertheless, this study finds a strong biological link between the bacteria and the fly. This apparent contradiction must be addressed in the revision.

Response to reviewer: Thank you for your valuable comments. We have rephrased the statements. And we have addressed the contradiction in discussion. See line 153-166. We hope that the revision will meet with approval.

Some references:

- Akami, M., Andongma, A. A., Chen, Z., Nan, J., Khaeso, K., Jurkevitch, E., . . . Yuval, B. (2019). (Diptera: Tephritidae). *Plos One*, 14(1). doi: 10.1371/journal.pone.0210109
- Diaz-Fleischer, F., Papaj, D. R., Prokopy, R. J., Norrbom, A. L., & Aluja, M. (2000). Evolution of fruit fly oviposition behavior. In M. Aluja & A. L. Norrbom (Eds.), *Fruit Flies (Tephritidae): Phylogeny and Evolution of Behavior* (pp. 811-841.). Boca Raton: CRC.
- Shu, R., Hahn, D. A., Jurkevitch, E., Liburd, O. E., Yuval, B., & Wong, A. C.-N. (2021). Sex-Dependent Effects of the Microbiome on Foraging and Locomotion in *Drosophila suzukii*. [Original Research]. *Frontiers in Microbiology*, 12(1094). doi: 10.3389/fmicb.2021.656406
- Silva, R., & Clarke, A. R. (2021). Aversive responses of Queensland fruit flies towards larval-infested fruits are modified by fruit quality and prior experience. *Journal of Insect Physiology*, 131. doi: 10.1016/j.jinsphys.2021.104231
- Sivakala, K. K., Jose, P. A., Shamir, M., Wong, A. C. N., Jurkevitch, E., & Yuval, B. (2022). Foraging behaviour of medfly larvae is affected by maternally transmitted and environmental bacteria. *Animal Behaviour*, 183, 169-176. doi: 10.1016/j.anbehav.2021.10.014
- Wong, A. C. N., Holmes, A., Ponton, F., Lihoreau, M., Wilson, K., Raubenheimer, D., & Simpson, S. J. (2015). Behavioral Microbiomics: A Multi-Dimensional Approach to Microbial Influence on Behavior. *Frontiers in Microbiology*, 6.
- Wong, A. C. N., Wang, Q. P., Morimoto, J., Senior, A. M., Lihoreau, M., Neely, G. G., . . . Ponton, F. (2017). Gut Microbiota Modifies Olfactory-Guided Microbial Preferences and Foraging Decisions in *Drosophila*. *Current Biology*, 27(15), 2397-+.

Response to reviewer: We have cited these studies in the revised manuscript.

Reviewers' comments:

Reviewer #1 (Remarks to the Author):

While the authors addressed several reviewer comments satisfactorily in the revised manuscript, some of the newly added or edited sections raise further questions or still don't answer some that should be addressed. Esp. added sections to initial questions on implications of aggregation induction by symbiosis by two reviewers resulted in a few added statements, still fail to integrate conflicting effects of different associated bacteria, competition of offspring and do not add a hypothesis or reasoning for the observation and some conclusions. Esp. the respective benefits for *Citrobacter* and *Bactrocera* and interaction between different associated microbes could be better laid out. I could imagine for *Bactrocera* in larvae possibly digestion or enrichment of dietary nutrients, defense against detrimental microbes (*Klebsiella* & *Providencia*) and adults ovary development. For *Citrobacter* transmission to new habitats by attracting oviposition and spreading with matured adults – do they successfully grow/persist on fruit without the presence of *Bactrocera*? Is there evidence for any of these in previous studies? See also following comments.

L159/160 Why is it beneficial for *Bactrocera* to oviposit into fruit already containing eggs even when infected with *Citrobacter* – larvae would later compete, as stated in L158.

L 161-166 I cannot follow the logic behind differential attraction/deterrence of ovary & egg associated bacteria *Citrobacter*, *Providencia* & *Klebsiella*. Why would *Citrobacter* colonize oviposition sites first or produce 3-HA before *Providencia* and *Klebsiella* colonize and release respective deterring cues? Is there data *Citrobacter* (initially) antagonizing growth of the other two bacteria, release volatiles at an earlier stage or a consistent succession of oviposited fruit colonization?

L172: Are there any indications how *Citrobacter* might support the development of *Bactrocera* offspring? This is important for the argumentation of evolution of stable transmission routes. Similarly, the benefit for *Citrobacter* should be included here.

Statistical test results are a crucial part of the results, thus, in my humble opinion, at least the most central tests of the results should be included in the main manuscript whenever significant differences are reported. The journal policy, read and confirmed by the authors actually supports this view in the reporting summary: "For all statistical analyses, confirm that the following items are present in the figure legend, table legend, main text, or Methods section. Line 8: For null hypothesis testing, the test statistic (e.g. F, t, r) with confidence intervals, effect sizes, degrees of freedom and P value noted"

Add test results for testing ANOVA assumptions.

L67 Remove "And"

L149 change "role of" to "to"

L160: change "contradict" to "a contradiction"

L161: change "study has" to "a previous study has"

L163 & 166: change "will be reproduced" to either "will reproduce" or "will have reproduced"

L253 Include at least a conceptual FISH protocol, esp. used probe(s) and coupled dye(s)

Fig1 Include negative control on Fig 1E / nested PCR Agarose gel

Reviewer #2 (Remarks to the Author):

In my opinion, the authors have successfully addressed the concerns raised. No further comments from my side

Reviewer #3 (Remarks to the Author):

I apologize for the tardiness in submitting my review.

The experimental part of the study, and its analysis, is very strong. Thanks to the revision, these sections are now much clearer.

In Lines 153-166 authors attempt to address the ecological context of their findings, as per my request. They write:

"Female insects will take many aspects into account to give their offspring the best chances of survival^{34,35}, and select a unique oviposition site for maximizing the survival, growth, and reproductive potential of the offspring³⁶. For *Bactrocera*, females do not oviposit preferentially in rotting fruit. Our previous study indicated that egg-surface bacteria (such as *Providencia* sp. and *Klebsiella* sp) can deter females from ovipositing¹⁷. Ovipositing in a fruit that is occupied by other larvae can be maladaptive for female flies, and indeed females prefer intact fruit over rotting fruit^{18,37}. However, this study indicates that females of *B. dorsalis* prefer to lay eggs in CF-BD infected fruits, which is contradict to the previous study. For *B. dorsalis*, study has shown that gravid female will usually lay dozens of eggs in one fruit³⁸. Thus, oviposition attraction or repellence may depend on the number of eggs and type of bacteria reproduced in fruit. At the initial stage, a small number of eggs were laid in fruits and the egg-surface *Citrobacter* will be reproduced first and attract oviposition of *B. dorsalis*. While at the later stage, large number of eggs were laid in fruit and the other egg-surface bacteria (such as *Providencia* sp. and *Klebsiella* sp) will be reproduced and repel oviposition."

First of all, this section can be written more clearly and with correct grammar, e.g.:

Female insects select oviposition sites that maximize the survival, growth, and reproductive potential of the offspring ^{34,35}³⁶. Ovipositing in a fruit that is occupied by other larvae can be maladaptive for female flies, and indeed *B. dorsalis* females prefer intact fruit over rotting fruit^{18,37}. Furthermore, our previous study indicated that egg-surface bacteria (such as *Providencia* sp. and *Klebsiella* sp) can deter females from ovipositing¹⁷.

However, the current study indicates that females of *B. dorsalis* prefer to lay eggs in CF-BD infected fruits, contradicts the previous study. *B. dorsalis* females will usually lay dozens of eggs in one fruit³⁸. Thus, oviposition attraction or repellence may depend on the number of eggs and type of bacteria reproduced in fruit. At the initial stage, a small number of eggs were laid in fruits and the egg-surface *Citrobacter* will be reproduced first and attract oviposition of *B. dorsalis*. While at the later stage, large number of eggs were laid in fruit and the other egg-surface bacteria (such as *Providencia* sp. and *Klebsiella* sp) will be reproduced and repel oviposition."

Secondly, there are still some logical problems with how this formulated. Why does "Thus, oviposition attraction or repellence may depend on the number of eggs and type of bacteria reproduced in fruit" follow from "*B. dorsalis* females will usually lay dozens of eggs in one fruit"? Authors ignore the findings of the reference they quote (38), which shows that clutch size varies greatly, according to host density and other factors. Indeed they (ref 38) found "female tendency to distribute eggs more widely and reduce the number of eggs laid per oviposition bout under high host density."

Perhaps then, this is the key to explain the apparent contradiction- when host density is low, repellence is adaptive, when host density is high, less so. Furthermore, and intriguingly, could there actually be an advantage in recruiting more oviposition attempts to a host? This is worth suggesting and may be the basis for more experiments in the future.

An alternative hypothesis is that CF-BD is present in the environment and establishes on suitable hosts, which become more attractive to oviposition as a result.

One minor comment: the sentence on lines 218-19: "This further suggests that the acquisition of beneficial bacteria by insects may be very efficient and complex" is not doing any work. It can be deleted.

We therefore invite you to revise and resubmit your manuscript, taking into account the points raised. In particular, the potential conflicting effects of different symbionts need to be addressed more clearly, interpretation of the results in the context of how symbionts are influencing oviposition and thus potentially competition of larvae as well as attraction and deterrence need to be addressed together with some other logical problems in the discussion.

Response: We have studied reviewers' comments carefully and have tried our best to revise our manuscript according to the reviewers' comments. According to the suggestions of reviewer 3, we have added some statements to address the potential conflicting effects of different symbionts.

Moreover, the statistical tests need to be integrated more precisely in the manuscript.

Response: Precise statistical test values were added in the figure legends.

Reviewers' comments:

Reviewer #1 (Remarks to the Author):

While the authors addressed several reviewer comments satisfactorily in the revised manuscript, some of the newly added or edited sections raise further questions or still don't answer some that should be addressed. Esp. added sections to initial questions on implications of aggregation induction by symbiosis by two reviewers resulted in a few added statements, still fail to integrate conflicting effects of different associated bacteria, competition of offspring and do not add a hypothesis or reasoning for the observation and some conclusions.

Response to reviewer: Thanks for your valuable comments. We are sorry for failure in addressing the concerns you have mentioned. To be honest, the conflicting effects of different bacteria also puzzled us, since we have no extract data to illustrate on what situation oviposition will be attracted. Thanks to reviewer 3, he has proposed some reasons for us. The first one is that host density may be the reason to explain the conflict effects of bacteria. For *B. dorsalis*, females will tend to distribute eggs more widely and reduce the number of eggs laid per oviposition bout under high host density. That means when host density is low, repellence is adaptive. The second one is that CF-BD may be present in the environment and establish on suitable hosts, which become more attractive to oviposition as a result. Of course we need conduct more experiments to verify such hypothesis in the future. We have revised the manuscript in discussion accordingly. Thanks again for your valuable comments. We hope our explanation and revision will meet with approval.

Esp. the respective benefits for *Citrobacter* and *Bactrocera* and interaction between different associated microbes could be better laid out. I could imagine for *Bactrocera* in larvae possibly digestion or enrichment of dietary nutrients, defense against detrimental microbes (*Klebsiella* & *Providencia*) and adults ovary development. For *Citrobacter* transmission to new habitats by attracting oviposition and spreading with matured adults

– do they successfully grow/persist on fruit without the presence of *Bactrocera*? Is there evidence for any of these in previous studies? See also following comments.

Response to reviewer: Thanks for your valuable comments. There are no much studies about the respective benefits for *Citrobacter* and *Bactrocera* and interaction between different associated microbes. However, some other studies give us some hints that *Citrobacter* may grow on fruit

without presence of *Bactrocera*. *Citrobacter* is ubiquitous in nature can be isolated from many fruits in orchards (Abadias, Canamas et al. 2006, Janisiewicz, Jurick et al. 2013, Adegun, Oluduro et al. 2019). And a number of studies have indicated that *Citrobacter* plays important nutrition supplying roles in both larva and adult of *Bactrocera* (Andongma, Wan et al. 2018, Rashid, Andongma et al. 2018, Hassan, Siddiqui et al. 2020). Considering your comments, we have added more information and quoted some more references in discussion to address such issues. Thanks again for your valuable comments. We hope the revision will meet with approval.

L159/160 Why is it beneficial for *Bactrocera* to oviposit into fruit already containing eggs even when infected with *Citrobacter* – larvae would later compete, as stated in L158.

Response to reviewer: Thanks for your comments. Indeed, we are not sure if it is beneficial for *Bactrocera* to oviposit into fruit already containing eggs. It is worth conducting more experiments in the future to investigate whether there actually is an advantage in recruiting more oviposition attempts to a host. Considering the suggestions of reviewer 3, we have proposed two explanations for the attraction and repellence in *Bactrocera*. We have proposed that host density may be one reason to explain the contradiction- when host density is low, repellence is adaptive. And an alternative hypothesis is that CF-BD may be present in the environment and establish on suitable hosts, which become more attractive to oviposition as a result. We hope the revision will meet with approval.

L 161-166 I cannot follow the logic behind differential attraction/deterrence of ovary & egg associated bacteria *Citrobacter*, *Providencia* & *Klebsiella*. Why would *Citrobacter* colonize oviposition sites first or produce 3-HA before *Providencia* and *Klebsiella* colonize and release respective deterring cues? Is there data *Citrobacter* (initially) antagonizing growth of the other two bacteria, release volatiles at an earlier stage or a consistent succession of oviposited fruit colonization?

Response to reviewer: Thanks for your question. To be honest, we cannot figure out why differential attraction/deterrence of ovary and egg associated bacteria (*Citrobacter*, *Providencia* & *Klebsiella*) exist in *Bactrocera*. And it is hard to collect the data to illustrate the questions you mention above. Considering the suggestion of reviewer 3, we have delete such statements and proposed two possible explanations for differential attraction/deterrence of the bacteria as the suggestion of reviewer 3.

L172: Are there any indications how *Citrobacter* might support the development of *Bactrocera* offspring? This is important for the argumentation of evolution of stable transmission routes. Similarly, the benefit for *Citrobacter* should be included here.

Response to reviewer: Thanks for your comments. Though there is no study reports how CF-BD supports the development of *Bactrocera* offspring, significant nutrition roles of *Citrobacter* in *Bactrocera* are shown by a number of studies. We have mentioned this in the revised manuscript and quoted the references. Thanks again for your comments. We hope the revision will meet with approval.

Statistical test results are a crucial part of the results, thus, in my humble opinion, at least the most central tests of the results should be included in the main manuscript whenever significant

differences are reported. The journal policy, read and confirmed by the authors actually supports this view in the reporting summary: “For all statistical analyses, confirm that the following items are present in the figure legend, table legend, main text, or Methods section. Line 8: For null hypothesis testing, the test statistic (e.g. F, t, r) with confidence intervals, effect sizes, degrees of freedom and P value noted”

Response to reviewer: Thanks for your valuable comments. We have added the detail statistical test results in the figure legends.

Add test results for testing ANOVA assumptions.

Response to reviewer: Thanks for your valuable comments. We have added the statistical test results in the figure legends.

L67 Remove “And”

Response to reviewer: done.

L149 change “role of” to “to”

Response to reviewer: done.

L160: change “contradict” to “a contradiction”

Response to reviewer: done.

L161: change “study has” to “a previous study has”

Response to reviewer: done.

L163 & 166: change “will be reproduced” to either “will reproduce” or “will have reproduced”

Response to reviewer: done.

L253 Include at least a conceptual FISH protocol, esp. used probe(s) and coupled dye(s)

Response to reviewer: done.

Fig1 Include negative control on Fig 1E / nested PCR Agarose gel

Response to reviewer: We have re-conducted the experiments to include the negative control.

Reviewer #3 (Remarks to the Author):

I apologize for the tardiness in submitting my review.

The experimental part of the study, and its analysis, is very strong. Thanks to the revision, these sections are now much clearer.

Response to reviewer: Thank you very much for your positive comments.

In Lines 153-166 authors attempt to address the ecological context of their findings, as per my request. They write:

“Female insects will take many aspects into account to give their offspring the best chances of survival^{34,35}, and select a unique oviposition site for maximizing the survival, growth, and reproductive potential of the offspring³⁶. For *Bactrocera*, females do not oviposit preferentially in

rotting fruit. Our previous study indicated that egg-surface bacteria (such as *Providencia* sp. and *Klebsiella* sp) can deter females from ovipositing¹⁷. Ovipositing in a fruit that is occupied by other larvae can be maladaptive for female flies, and indeed females prefer intact fruit over rotting fruit^{18,37}. However, this study indicates that females of *B. dorsalis* prefer to lay eggs in CF-BD infected fruits, which is contradict to the previous study. For *B. dorsalis*, study has shown that gravid female will usually lay dozens of eggs in one fruit³⁸. Thus, oviposition attraction or repellence may depend on the number of eggs and type of bacteria reproduced in fruit. At the initial stage, a small number of eggs were laid in fruits and the egg-surface *Citrobacter* will be reproduced first and attract oviposition of *B. dorsalis*. While at the later stage, large number of eggs were laid in fruit and the other egg-surface bacteria (such as *Providencia* sp. and *Klebsiella* sp) will be reproduced and repel oviposition.”

First of all, this section can be written more clearly and with correct grammar, e.g.:

Female insects select oviposition sites that maximize the survival, growth, and reproductive potential of the offspring ^{34,35}36. Ovipositing in a fruit that is occupied by other larvae can be maladaptive for female flies, and indeed *B. dorsalis* females prefer intact fruit over rotting fruit^{18,37}. Furthermore, our previous study indicated that egg-surface bacteria (such as *Providencia* sp. and *Klebsiella* sp) can deter females from ovipositing¹⁷.

However, the current study indicates that females of *B. dorsalis* prefer to lay eggs in CF-BD infected fruits, contradicts the previous study. *B. dorsalis* females will usually lay dozens of eggs in one fruit³⁸. Thus, oviposition attraction or repellence may depend on the number of eggs and type of bacteria reproduced in fruit. At the initial stage, a small number of eggs were laid in fruits and the egg-surface *Citrobacter* will be reproduced first and attract oviposition of *B. dorsalis*. While at the later stage, large number of eggs were laid in fruit and the other egg-surface bacteria (such as *Providencia* sp. and *Klebsiella* sp) will be reproduced and repel oviposition.”

Secondly, there are still some logical problems with how this formulated. Why does “Thus, oviposition attraction or repellence may depend on the number of eggs and type of bacteria reproduced in fruit” follow from “*B. dorsalis* females will usually lay dozens of eggs in one fruit”? Authors ignore the findings of the reference they quote (38), which shows that clutch size varies greatly, according to host density and other factors. Indeed they (ref 38) found “female tendency to distribute eggs more widely and reduce the number of eggs laid per oviposition bout under high host density.”

Perhaps then, this is the key to explain the apparent contradiction- when host density is low, repellence is adaptive, when host density is high, less so. Furthermore, and intriguingly, could there actually be an advantage in recruiting more oviposition attempts to a host? This is worth suggesting and may be the basis for more experiments in the future.

An alternative hypothesis is that CF-BD is present in the environment and establishes on suitable hosts, which become more attractive to oviposition as a result.

Response to reviewer: Thank you very much for your suggestions in revising this part. We have revised the manuscript according to your constructive suggestions. Thanks again. We hope the revision will meet with approval.

One minor comment: the sentence on lines 218-19: “This further suggests that the acquisition of beneficial bacteria by insects may be very efficient and complex” is not doing any work. It can be deleted.

Response to reviewer: done.

REVIEWERS' COMMENTS:

Reviewer #1 (Remarks to the Author):

Thank you for the repeated revision of the statistical results and discussion. Especially the revised discussion addresses crucial aspects now appropriately and also indicates which questions need to be addressed by future work.

Some minor comments:

L 173, 179 and 182: Please remove/replace "And..." at the beginning of these sentences.

Hannes Schuler (editorial board member) (Remarks to the Author):

1. All the outcome of the statistical tests claimed by reviewer 1 are included only in the figure legends. This should be included also in the results.

2. In some parts the new paragraphs (highlighted in yellow), there are some mistakes:

-L162: Please make a ':' instead of a '.'

-L165: Fruits in plural

-L166: 'and the bacteria they bring' is unclear

-169: 'It's worth conducting more experiments' sounds a bit slangy

-L179, L182: Sentences beginn 2x with 'and'

REVIEWERS' COMMENTS:

Reviewer #1 (Remarks to the Author):

Thank you for the repeated revision of the statistical results and discussion. Especially the revised discussion addresses crucial aspects now appropriately and also indicates which questions need to be addressed by future work.

Some minor comments:

L 173, 179 and 182: Please remove/replace "And..." at the beginning of these sentences.

Response: Done.

Hannes Schuler (editorial board member) (Remarks to the Author):

1. All the outcome of the statistical tests claimed by reviewer 1 are included only in the figure legends. This should be included also in the results.

Response: Done.

2. In some parts the new paragraphs (highlighted in yellow), there are some mistakes:

-L162: Please make a ':' instead of a '.'

Response: Done.

-L165: Fruits in plural

Response: done.

-L166: 'and the bacteria they bring' is unclear

Response: Corrected.

-169: 'It's worth conducting more experiments' sounds a bit slangy

Response: Corrected.

-L179, L182: Sentences beginn 2x with 'and'

Response: Deleted.